

# Pollinator diversity and reproductive success of *Epipactis helleborine* (L.) Crantz (Orchidaceae) in anthropogenic and natural habitats

Agnieszka Rewicz[1], Radomir Jaskuła[2], Tomasz Rewicz[3] and Grzegorz Tończyk[2]

[1] Department of Geobotany and Plant Ecology, University of Lodz, Łódź, Poland
[2] Department of Invertebrate Zoology and Hydrobiology, University of Lodz, Łódź, Poland
[3] Laboratory of Microscopic Imaging and Specialized Biological Techniques, University of Lodz, Łódź, Poland

Corresponding author
Agnieszka Rewicz,
agnieszka.rewicz@biol.uni.lodz.pl,
stefa@biol.uni.lodz.pl

## ABSTRACT

**Background.** *Epipactis helleborine* is an Eurasian orchid species which prefers woodland environments but it may also spontaneously and successfully colonise human-made artificial and disturbed habitats such as roadsides, town parks and gardens. It is suggested that orchids colonising anthropogenic habitats are characterised by a specific set of features (e.g., large plant size, fast flower production). However, as it is not well known how pollinator diversity and reproductive success of *E. helleborine* differs in populations in anthropogenic habitats compared to populations from natural habitats, we wanted to compare pollinator diversity and reproductive success of this orchid species between natural and anthropogenic habitat types.

**Methods.** Pollination biology, reproductive success and autogamy in populations of *E. helleborine* from anthropogenic (roadside) and natural (forest) habitats were compared. Eight populations (four natural and four human-disturbed ones) in two seasons were studied according to height of plants, length of inflorescences, as well as numbers of juvenile shoots, flowering shoots, flowers, and fruits. The number and diversity of insect pollinators were studied in one natural and two human-disturbed populations.

**Results.** Reproductive success (the ratio of the number of flowers to the number of fruits) in the populations from anthropogenic habitats was significantly higher than in the natural habitats. Moreover, plants from anthropogenic habitats were larger than those from natural ones. In both types of populations, the main insect pollinators were Syrphidae, Culicidae, Vespidae, Apidae and Formicidae. With respect to the type of pollinators' mouth-parts, chewing (39%), sponging (34%) and chewing-sucking (20%) pollinators prevailed in anthropogenic habitats. In natural habitats, pollinators with sponging (55%) and chewing mouth-parts (32%) dominated, while chewing-sucking and piercing-sucking insects accounted for 9% and 4% respectively.

**Discussion.** We suggest that higher reproductive success of *E. helleborine* in the populations from anthropogenic habitats than in the populations from natural habitats may result from a higher number of visits by pollinators and their greater species diversity, but also from the larger size of plants growing in such habitats. Moreover, our data clearly show that *E. helleborine* is an opportunistic species with respect to pollinators, with a wide spectrum of pollinating insects. Summarising, *E. helleborine* is a rare example of orchid species whose current range is not declining. Its ability to

make use of anthropogenically altered habitats has allowed its significant spatial range expansion, and even successful colonisation of North America.

## INTRODUCTION

Orchidaceae is one of the most diverse and species-rich (20,000–30,000 species) plant families (*Baumann, Kunkele & Lorenz, 2010*; *Djordjević et al., 2016a*), with many species that are seriously endangered and require conservation efforts to maintain their populations.

Destruction of natural habitats is causing extinction of many orchid species (*Swarts & Dixon, 2009*). However, some orchid species, especially in temperate regions of Europe and North America, found anthropogenic habitats as suitable as natural ones (*Pedersen, Watthana & Srimuang, 2013*). A recent study of orchids in Turkey has indicated that graveyards are places where orchid species occur frequently (*Löki et al., 2015*). In addition, *Djordjević et al. (2016b)* have noted that *Himantoglossum calcaratum*, *Anacamptis pyramidalis* and *Ophrys* species often grow in habitats along the roads in western Serbia. Moreover, the same authors have shown that *Orchis purpurea* can be an indicator of ruderal habitat type. The most common colonisers of secondary habitats in Central Europe are *Epipactis* and *Dactylorhiza* species (*Adamowski, 2004*; *Adamowski, 2006*; *Esfeld et al., 2008*; *Rewicz, Kołodziejek & Jakubska-Busse, 2016*). Moreover, *Jurkiewicz et al. (2001)* noted that populations of *Epipactis atrorubens*, *E. helleborine* and *Dactylorhiza majalis* were observed on several mine tailings in southern Poland. Colonisation of such habitats is fuelled by disturbances of surface soil layers and exposure of deeper layers including bedrock (i.e., quarries where limestone or chalk are exposed). Additionally, the surface layer of soil and vegetation could be destroyed, which weakens the competitive potential of original vegetation (*Adamowski, 2006*). Orchids that colonize anthropogenic habitats are characterized by a specific set of features: fast growth resulting in large plant size and fast flower production (*Forman et al., 2009*).

An important aspect of orchid population biology is its unique reproductive system (*Machaka-Houri et al., 2012*), which in this case means mass production of very small and light seeds (from 0.31 µg to 24 µg, depending on the species) (*Arditti, 1967*). However, the high number of seeds does not lead to high recruitment of seedlings. Low reproductive success, which is defined as the ratio of the number of flowers to the number of fruits (*Doust & Doust, 1988*), may be caused by high level of morphological adaptation of flowers to particular pollinators. More than 60% of all orchid species are pollinated by a single pollinator (*Tremblay, 1992*; *Tremblay et al., 2005*).

Human disturbance, especially habitat transformation (including its impact on soil, moisture conditions, and changes of the floristic composition of plant communities), is regarded as a principal cause of pollinator decline on a global scale (*Goulson, Lye & Darvill,*

*2008*), as well as an important factor directly and indirectly changing the species structure of pollinating insects (*Clemente, 2009*). Deficiency of suitable pollinators may also be a reason for low reproductive success. Thus, autogamy (self-autonomy) is an alternative way of seed production. A question arises which reproductive system is preferred by orchids rapidly and successfully colonising anthropogenic habitats (*Light & MacConaill, 2006*). *Epipactis helleborine* can be a suitable model species, as it occurs in both natural and disturbed habitats, is able to undergo both auto- and allogamy and, furthermore, has been successfully established in North America. *Ehlers, Olesen & Gren (2002)* have investigated the reproductive success of *E. helleborine* in natural habitats; however, there is a lack of knowledge concerning the reproductive success and pollinator diversity of *E. helleborine* in relation to certain types of natural and anthropogenic habitats. Moreover, in the literature there are no detailed data about the diversity of insect pollinators of *E. helleborine* in relation to the type of their mouth-parts. Such information is important as some types of mouth-parts allow insects to collect food from morphologically different types of flowers, as a result such insects can be pollinators of many taxonomically different plant groups. In addition, insects with specialised mouth-parts can collect food only from one or two morphologically different types of flowers, which means that they can only be pollinators of a narrow number of plants (*Gillott, 2005*). Some orchid species have flowers which are morphologically adapted for nectar and pollen collecting by insects with different types of mouth-parts, while other can be pollinated only by insects with one, sometimes strongly specialised, type of mouth-parts.

In this study, we compared the reproductive success and pollinator diversity of *E. helleborine* from natural and anthropogenic habitats. Specifically, we addressed the following questions: (a) what is the composition of and the differences in the pollinator fauna of *E. helleborine* from anthropogenic and natural habitats in terms of insect diversity and diversity of insect mouth-parts; (b) does the number of capsules produced through autogamy and natural pollination differ between populations from anthropogenic and natural habitats; (c) is the reproductive success of *E. helleborine* different in populations from anthropogenic and natural habitats?

## MATERIALS AND METHODS

### Studied species and study area

*Epipactis helleborine* (L.) Crantz (broad-leaved helleborine) is a Eurasian species (*Kolanowska, 2013*) introduced in the 19th century (*Owen, 1879*) to several regions of North America (*Procházka & Velísek, 1983*). It usually grows in deciduous and coniferous forest communities, on edges and in clearings in woodland, up to 2,000 m a.s.l (*Delforge, 2006*). Furthermore, this species inhabits different types of anthropogenic habitats such as: roadsides, cemeteries, poplar plantations, gravel pits, quarries, railway embankments, and mine tailings (*Świercz, 2004*; *Świercz, 2006*; *Kiedrzyński & Stefaniak, 2011*), and may also appear spontaneously in urban areas such as town parks and gardens (*Kolanowska, 2013*). Some studies have shown that *E. helleborine* is a species with a broad ecological tolerance, which is not highly specialised, and often acts as a pioneer in human-disturbed
areas (*Piękoś-Mirkowa & Mirek, 2006*; *Tsiftsis et al., 2008*). *Epipactis helleborine* is thus a fine example of apophytism—i.e., a native species growing in disturbed or human-made habitats (*Sukopp, 2006*; *Rewicz et al., 2015*). Orchids that colonise anthropogenic habitats are characterised by a specific set of features: fast growth resulting in large plant size and fast flower production (*Forman et al., 2009*).

We accepted the definition of reproductive success after *Doust & Doust (1988)* as the ratio of the number of flowers to the number of fruits. All shoots found in each population were measured. The following parameters were recorded for each plant in a population: number of juvenile shoots (JS), number of flowering shoots (NFS), number of flowers (NF), number of fruits (capsules) (NFR), and height of plants (HP). Population size includes the total number of shoots (juvenile and flowering). The area occupied by a given population was determined by the most remote ramets of *E. helleborine*. Border posts indicating the surface area of the population were placed at intervals of one meter from each other, and then the edges of the study area were delineated using a piece of string. The population area measured using a square net was approximately 1 m$^2$ (Table 1). Density of each population was also measured as shoots/m$^2$. The consent to conduct the research was issued by the Regional Directorate of Environmental Protection in Białystok, permit no. WPN6400.74.2013.MW and the Ministry of the Environment—permit no. 35/17258/12/RS.

Eight populations of *E. helleborine* occurring in Poland were studied in two seasons, 2011 and 2012 (July—flowering and pollinators, September—seed capsules collection). The identified habitat types were divided in to two categories. One included human-disturbed habitats, such as roadsides (population A1—between a road and a wooden fence in the village of Guszczewina; A2—close to a car park in Hajnówka; A3—in a thicket by a roadside in Sulejów; A4—on a roadside bordering *Pinus sylvestris* forest in Sulejów) (Fig. 1). The other one grouped natural habitats: population N1—in a forest of *Galio sylvatici-Carpinetum betuli* Oberd. 1957 in Kotowice (*Jakubska & Orlowski, 2003*); N2—in a forest of *Galio sylvatici-Carpinetum betuli* Oberd. 1957 in Kaczawskie Mts. (*Kwiatkowski, 2006*); N3, N4—*Galio sylvatici-Carpinetum betuli* Oberd. 1957 in the Strict Reserve of the Białowieża Primeval Forest (*Faliński, 2001*) (Fig. 1, Table 1).

The detailed review of the vascular flora of the studied habitats (the sample area was 30 m$^2$) is presented in Table S1. For each species recorded in this study, its relative frequency of occurrence was calculated (*Brower & Zar, 1984*). Frequency (*Fi*) was calculated according to the formula: $Fi = (j_i/k) \times 100\%$, where: $j$—the number of populations in which species '*i*' was recorded, and $k$—the total number of populations (Table S1).

In total, 68 species of vascular plant species were recorded; 38 (ten species of trees) in natural and 38 (five species of trees) in anthropogenic habitats. The most frequently occurring plants in natural habitats were: *Carpinus betulus*, *Veronica chamaedrys* and *Rubus* sp., registered in all the study plots (*Fi* = 100%). The most frequently occurring plants in populations in anthropogenic habitats were: *Achillea millefolium*, *Conyza canadensis*, *Dactylis glomerata*, *Galium aparine*, *Medicago lupulina*, *Poa annua*, *P. nemoralis* and *Vicia cracca*.

**Table 1** **Studied populations of *Epipactis helleborine*.**

| Population code | Locality | Coordinates | Altitude (m asl) | Area (m²) | Population size (number of shoots) | Density (shoots/m²) |
|---|---|---|---|---|---|---|
| **Anthropogenic habitats** | | | | | | |
| A1[*] | roadside (Guszczewina) | N52.831600 E23.794836 | 148 | 36 | 127 | 3.53 |
| A2[*] | roadside (Hajnówka) | N52.734217 E23.603314 | 181 | 108 | 102 | 0.94 |
| A3 | roadside (Sulejów) | N51.353793 E19.883155 | 166 | 460 | 80 | 0.17 |
| A4 | roadside (Sulejów) | N51.349757 E19.882484 | 167 | 46 | 152 | 3.30 |
| **Natural habitats** | | | | | | |
| N1 | mixed forest (Kotowice) | N51.041241 E17.176701 | 128 | 100 | 300 | 3.00 |
| N2 | mixed forest (Kaczawskie Mts) | N50.963255 E15.963255 | 480 | 40 | 150 | 3.75 |
| N3[*] | mixed forest (Białowieża Primeval Forest) | N52.800743 E23.914125 | 51 | 120 | 34 | 0.28 |
| N4 | mixed forest (Białowieża Primeval Forest) | N52.7650022 E23.884316 | 44 | 400 | 41 | 0.10 |

**Notes.**

[*]Asterix indicates populations where pollinator fauna were analyzed.

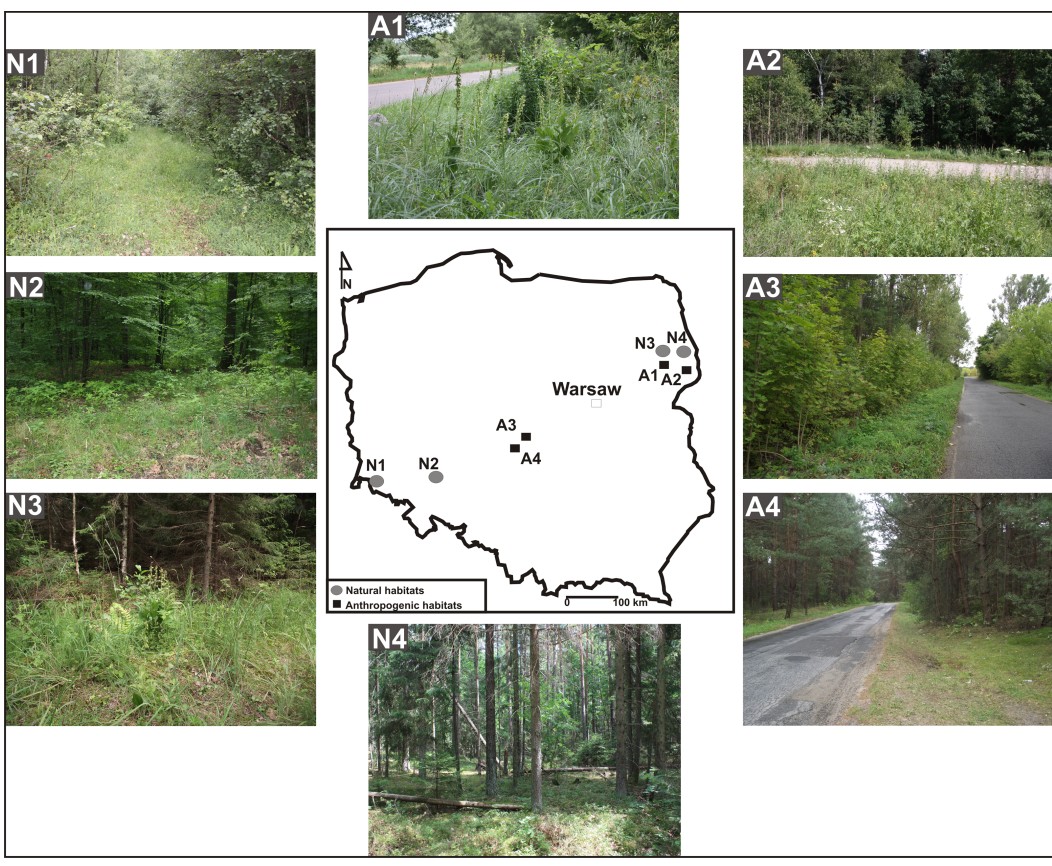

**Figure 1** **Habitats of *E. helleborine*.** A —anthropogenic habitat, N—natural habitat, A1—between a road and a wooden fence in the village of Guszczewina, A2—close to a car park in Hajnówka, A3—in a thicket by a roadside in Sulejów, A4— on a roadside bordering a *Pinus sylvestris* forest in Sulejów, N1—a forest of *Galio sylvatici-Carpinetum betuli* Oberd. 1957 in Kotowice, N2—a forest of *Galio sylvatici-Carpinetum betuli* Oberd. 1957 in Kaczawskie Mts., N3 and N4—*Galio sylvatici-Carpinetum betuli* Oberd. 1957 in the natural habitat (Strict Reserve of the Białowieża Primeval Forest).

## Pollinators of *E. helleborine*

Pollinators were both caught and observed in two populations of *E. helleborine* in two anthropogenic habitats: a roadside in the village of Guszczewina (A1), and close to a car parking in the city of Hajnówka (A2), and in one population in a natural habitat—in the Białowieża Primeval Forest (N3) (Fig. 1, Table 1).

Insects from fresh orchid flowers were caught using entomological hand nets in two periods: 15.–23.07.2011 and 13.–22.07.2012. The fieldwork was carried out during days with sunny weather. In the anthropogenic (A1, A2) *E. helleborine* populations, the insects were caught by two two-people teams between 9 a.m. and 7 p.m. In the population (N3) from natural habitat, insects were caught by four people to get comparable hours effort between populations from anthropogenic and natural habitat. Insects were collected from 10 shoots (20 in the natural habitat) growing close to each other. Specimens were collected until the transfer of pollinia by pollinators was observed. Insects were killed using ethyl acetate and preserved in 75% ethanol (except bumble-bees *Bombus* spp. which are

protected by Polish law—these specimens were only photographed). Insects which arrived with attached pollinia or departed the flower with pollinia were recognised as pollinators. The number of captured insects corresponded to the number of visits, with the exception of the *Bombus* species, for which only the number of visits was counted. Identification of the insects to order/family levels was done based on *Gillott (2005)*, *Hůrka (2005)* and *Oosterbroek (2006)*, while entomological nomenclature of insect mouth-parts was provided according to *Gillott (2005)*.

### The ability of *E. helleborine* to undergo autogamy

The autogamy experiment was carried out from July to September 2012. Ten shoots in the early stage of flowering (closed buds) were selected in each population for the autogamy experiment. Flowers on each shoot were counted and inflorescences were covered by bags made from a mosquito net. We also used ten control plants (not covered by mosquito-net bags). After three months, the isolators were removed and the number of fruit sets was counted. Viability of seed was examined by the tetrazolium test (live seed with stained embryos and dead seed with unstained embryos) (*Van Waes & Debergh, 1986*).

### Data analysis

The software package STATISTICA PL. ver. 10 (*StatSoft Inc, 2011*) was used for all the statistical analyses (*Van Emden, 2008*). Diversity of pollinator fauna in natural and anthropogenic habitats was evaluated using the chi-squared test. To compare reproductive success between habitats, we used the Student's $t$-test (we used values of individuals). Correlation between the number of flowers and the number of fruits (we used average) in inflorescence in different habitats was evaluated using the Spearman's correlation The relationship between reproductive success and the height of plant and number of flowers was investigated by linear regression model (*Meissner, 2010*). To compare the number of fruits (capsules) produced by autogamy in different habitats (we used average) we used the Mann–Whitney $U$ test.

## RESULTS

### Pollinators of *E. helleborine*

Pollinators of *E. helleborine* collected during this study belonged to six orders and 24 families of insects (Table 2, Fig. 2). In the case of the populations from the anthropogenic habitats, taxonomic diversity of pollinators was higher, with 19 families grouped in five orders, while in the natural habitats we noted only 14 families from four orders (statistically significant values, chi-squared test, $\chi^2 = 0.001161$, $df = 6$, $p = 0.05$). In the population from the natural habitat, the most frequent families were Syrphidae (111 visits) and Vespidae (44). In the populations from the anthropogenic habitats, the most frequent families were Syrphidae (57), Vespidae (48) and Apidae (43). In both types of habitats, Diptera and Hymenoptera clearly dominated, with 41% and 52% of all the pollinators observed in the populations from the anthropogenic habitats, and with 59% and 37% observed in the population from the natural habitat (Fig. 3). Coleoptera were the third main group of pollinators making up 6% of the populations from the anthropogenic habitats and 4% of

Peer J

**Table 2  Pollinators of *Epipactis helleborine* in natural and anthropogenic habitats.**

| Taxon of pollinator | | Type of mouthparts | Habitat type | |
|---|---|---|---|---|
| Order | Family | | Natural | Anthropogenic |
| Orthoptera | Acrididae | C | 0 | 1 |
| Dermaptera | Forficulidae | C | 0 | 1 |
| Diptera | Calliphoridae | S | 1 | 7 |
| | Culicidae | PS | 9 | 16 |
| | Lauxanidae | S | 0 | 1 |
| | Muscidae | S | 1 | 3 |
| | Scathopagidae | S | 0 | 1 |
| | Sepsidae | S | 0 | 1 |
| | Syrphidae | S | 111 | 57 |
| | Tachinidae | S | 1 | 2 |
| | Tephritidae | S | 0 | 1 |
| | Tipulidae | S | 0 | 1 |
| Mecoptera | Planorpidae | C | 4 | 0 |
| Hymenoptera | Apidae | CS | 18 | 43 |
| | Formicidae | C | 13 | 18 |
| | Ichneumonidae | C | 1 | 4 |
| | Pamphiliidae | C | 0 | 1 |
| | Vespidae | C | 44 | 48 |
| Coleoptera | Cantharidae | C | 0 | 10 |
| | Cerambycidae | C | 1 | 0 |
| | Coccinellidae | C | 0 | 4 |
| | Elateridae | C | 2 | 0 |
| | Melyridae | C | 4 | 0 |
| | Nitidulidae | C | 1 | 0 |
| Total | | | 208 | 220 |

Notes.
C, chewing; S, sponging; PS, piercing and sucking; CS, chewing and sucking.

the population from the natural habitat. Occasionally, single individuals of grasshoppers (Orthoptera) and earwigs (Dermaptera) were also noted as pollinators of *E. helleborine* in the populations from the anthropogenic habitats and scorpion flies (Mecoptera) in the population from the natural habitat.

In the populations from the anthropogenic habitats, the main dipteran pollinators were hoverflies (Syrphidae), making up 63% of dipteran pollinators and 26% of all the observed pollinators, with the most frequent species *Meliscaeva cinctella* and *Episyrphus balteatus*, followed by mosquitoes (Culicidae) (18% of dipteran and 7% of all the pollinators). The main hymenopteran pollinators were wasps (Vespidae—42% of hymenopteran and 22% of all the pollinators, with the most frequent species *Dolichovespula saxonica*), bees (Apidae—38% and 20%, respectively, with the main pollinator *Apis mellifera*), and ants (Formicidae—16% and 8% respectively) (Fig. 3). In the population from the natural habitat, true flies (Syrphidae) accounted for 90% (53% of all the pollinators; with *Meliscaeva cinctella* as the most frequent species), and mosquitoes (Culicidae) made up
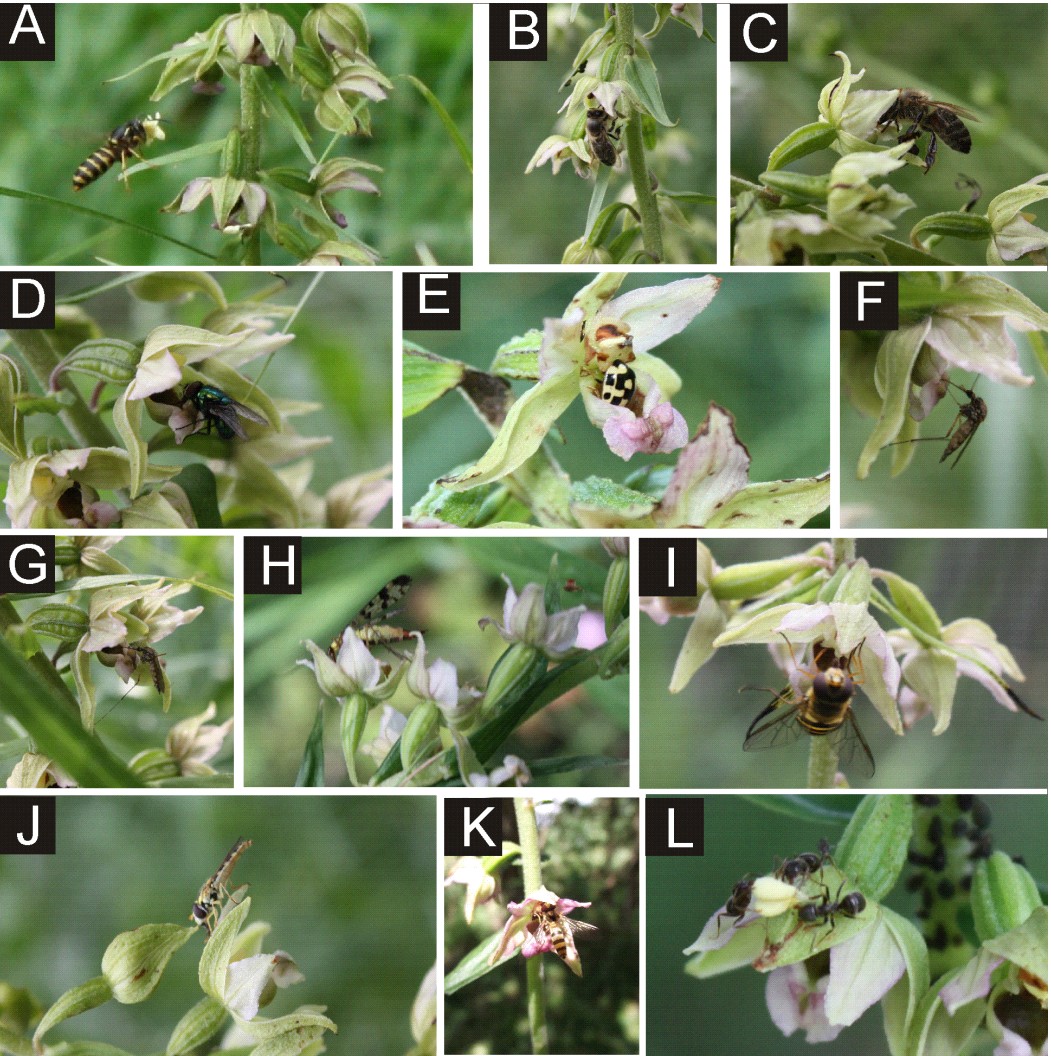

**Figure 2** **Pollinators of *E. helleborine*.** (A) wasp (Vespidae) with pollinia attached to its head, (B–C) –honeybees (Apidae), (D) carrion fly (Calliphoridae), (E) ladybird (Coccinellidae), (F–G) mosquito (Culicidae), (H) scorpionfly (Panorpidae), (I–K) hoverflies (Syrphidae), (L) ants (Formicidae) (photo: A Rewicz 2011/2012).

7% of pollinators (4% of all the pollinators), while the main hymenopteran pollinators were wasps (Vespidae—58% of hymenopterans and 21% of all the pollinators; with *Dolichovespula saxonica* as the most frequent pollinator), bees (Apidae—24% and 9%, respectively), and ants (Formicidae—17% and 6%, respectively) (Fig. 3).

According to the type of mouth-parts, the pollinators of *E. helleborine* can be ascribed to four groups: 1/sponging insects (Diptera excluding Culicidae—44% of all the noted pollinators), 2/ chewing (= mandibulate) insects (Hymenoptera excluding Apidae, Coleoptera, Dermaptera, Orthoptera and Mecoptera—36%), 3/chewing-sucking insects (Apidae—14%), and 4/piercing and sucking insects (Culicidae—6%) (Fig. 4).

In the population from the natural habitat, the main groups of pollinators of *E. helleborine* were sponging (55%) and chewing insects (32%), while the chewing-sucking

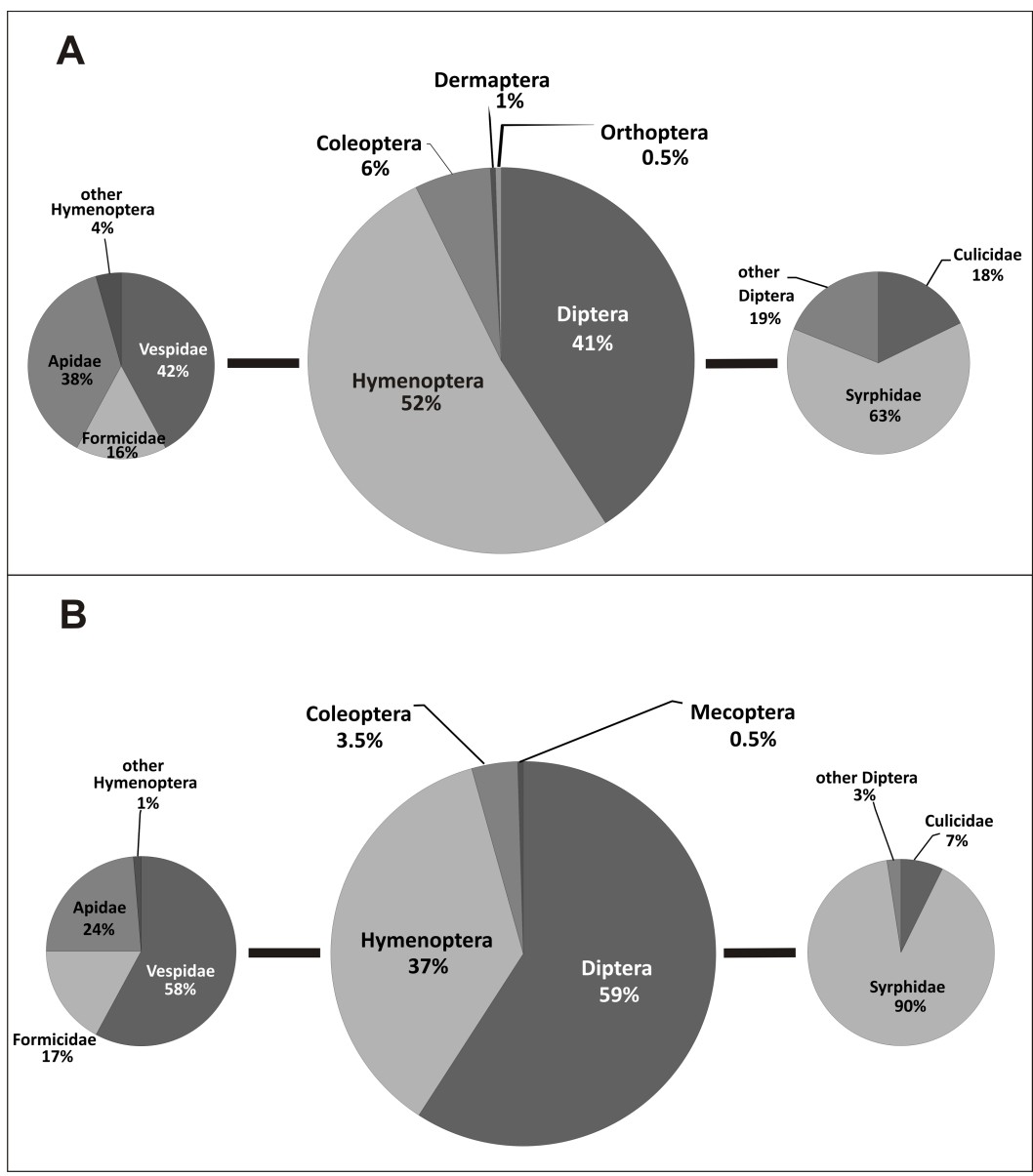

**Figure 3** **Taxonomic diversity of *E. helleborine* pollinators in anthropogenic (A) and natural (B) habitats.**

and piercing and sucking insects made up respectively 9% and 4% of all the pollinators. In the populations from the anthropogenic habitats, the most frequent pollinators belonged to the groups of chewing (39%), sponging (34%), and chewing-sucking (20%) insects, and only 7% of the noted insects were characterised by piercing and sucking mouth-parts.

## Reproductive success

Reproductive success in the populations from anthropogenic habitats (average from 2011 to 2012—87.1%) was significantly higher than in the populations from the natural habitats (average from 2011 to 2012—72.3%) (Student's $t$-test, $p = 0.02$, $df = 14$). In the
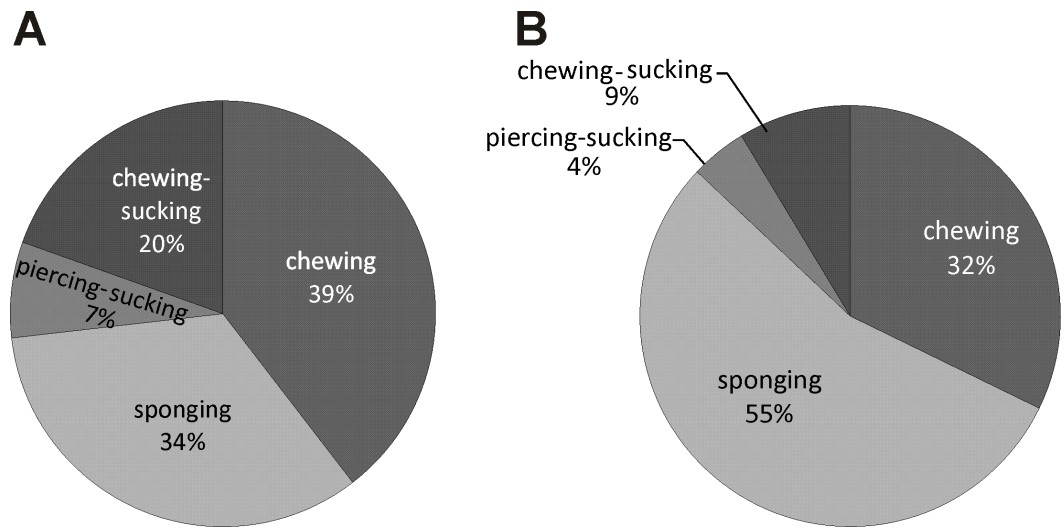

**Figure 4** Diversity of *E. helleborine* pollinators in anthropogenic (A) and natural (B) habitats based on insect mouthparts.

populations from the anthropogenic habitats, reproductive success ranged from 77.8% (A4—in 2011) to 100% (A2—2012), while in the populations from the natural habitats, it ranged from 44.4% (N1—2011) to 85.0.3% (N1—2012) (Table 3). The number of flowers in the populations from the natural habitats ranged from 20 (in 2011) to 22 (in 2012), while in the populations from the anthropogenic habitats from 15 (2011) to 14 (2012). Differences between habitats of the number of flowers were statistically significant (Student's $t$-test, $p = 0.03$).

The strongest correlation was found between the reproductive success and height of plants ($r = 0.82$, $p < 0.05$) in the populations from the anthropogenic habitats. No significant correlation was found between reproductive success and population density. In the populations from natural habitats, a weak correlation was found between reproductive success and density of populations ($r = 0.40$, $p < 0.05$). The regression analysis was significant only in the case of populations from the anthropogenic habitats between reproductive success and height of plants, as well as between reproductive success and the number of flowers (Fig. 5).

### Autogamy

The mean number of capsules (20) produced in the autogamy treatment in the populations from the anthropogenic habitats was significantly higher than the number of capsules produced in the population from the natural habitat (12 capsules) (Mann–Whitney $U$ test, $Z = 3.30$, $p = 0.0008$). In the case of autogamy in the populations from the natural and anthropogenic habitats, the number of capsules was strongly positively correlated with the number of flowers per inflorescence (Spearman's correlation, $r = 88$, $p < 0.05$) (Fig. 6). In the case of natural pollination, in both the populations from anthropogenic and natural habitats, the number of fruits was the same (19) and the number of capsules was

**Table 3** **Reproductive success of *Epipactis helleborine* in natural and anthropogenic habitats.**

| Site | D (n/m$^2$) | HP | NFS | NF | NFR | RS | Site | D (n/m$^2$) | HP | NFS | NF | NFR | RS |
|---|---|---|---|---|---|---|---|---|---|---|---|---|---|
| | | **Anthropogenic habitats** | | | | | | | **Natural habitats** | | | | |
| | | | **2011** | | | | | | | **2011** | | | |
| A1 | 0.28 | 84.79 ± 24.04 | 39 | 20 ± 9.20 | 19 ± 9.48 | 95.00 ± 7.20 | N1 | 0.83 | 61.96 ± 17.91 | 25 | 18 ± 11.8 | 8 ± 7.80 | 44.40 ± 5.13 |
| A2 | 0.93 | 54.08 ± 15.84 | 26 | 19 ± 8.63 | 16 ± 7.83 | 84.20 ± 8.30 | N2 | 0.77 | 57.36 ± 15.99 | 30 | 18 ± 10.8 | 13 ± 8.36 | 72.20 ± 8.78 |
| A3 | 0.9 | 56.29 ± 11.80 | 22 | 12 ± 6.20 | 10 ± 5.92 | 83.30 ± 10.4 | N3 | 0.21 | 34.53 ± 15.27 | 28 | 18 ± 12.5 | 14 ± 11.99 | 77.80 ± 9.54 |
| A4 | 0.4 | 41.98 ± 12.64 | 38 | 9 ± 9.25 | 7 ± 9.21 | 77.80 ± 12.34 | N4 | 0.88 | 48.36 ± 23.60 | 33 | 24 ± 14.8 | 20 ± 13.15 | 83.30 ± 11.40 |
| Average | 0.63 | 59.28 | 31.25 | 15 | 13 | 85.08 | Average | 0.67 | 50.55 | 29 | 20 | 14 | 69.42 |
| | | | **2012** | | | | | | | **2012** | | | |
| A1 | 0.38 | 87.65 ± 22.48 | 43 | 19 ± 8.37 | 18 ± 8.28 | 94.70 ± 6.54 | N1 | 1.24 | 66.67 ± 13.60 | 15 | 20 ± 6.34 | 17 ± 5.19 | 85.00 ± 7.45 |
| A2 | 0.77 | 64.3 ± 19.51 | 45 | 20 ± 14.1 | 20 ± 13.87 | 100.00 ± 10.30 | N2 | 0.97 | 62.15 ± 16.58 | 24 | 19 ± 11.4 | 14 ± 10.00 | 73.70 ± 6.78 |
| A3 | 2.39 | 56.0 ± 19.07 | 13 | 6 ± 6.20 | 5 ± 5.81 | 83.30 ± 8.56 | N3 | 0.22 | 55.22 ± 20.32 | 25 | 24 ± 13.4 | 15 ± 10.45 | 62.50 ± 5.45 |
| A4 | 0.33 | 40.24 ± 15.97 | 46 | 9 ± 8.73 | 7 ± 8.66 | 77.80 ± 8.34 | N4 | 0.81 | 46.07 ± 23.91 | 32 | 25 ± 14.3 | 20 ± 14.00 | 80.00 ± 9.45 |
| Average | 0.97 | 62.05 | 36.75 | 14 | 12 | 88.95 | Average | 0.81 | 57.53 | 24 | 22 | 17 | 75.30 |

**Notes.**

D(n/m$^2$), density of population; HP, height of plants; NFS, number of flowering shoots; NF, number of flowers; NFR, number of fruits; RS, reproductive success, ±standard deviation.

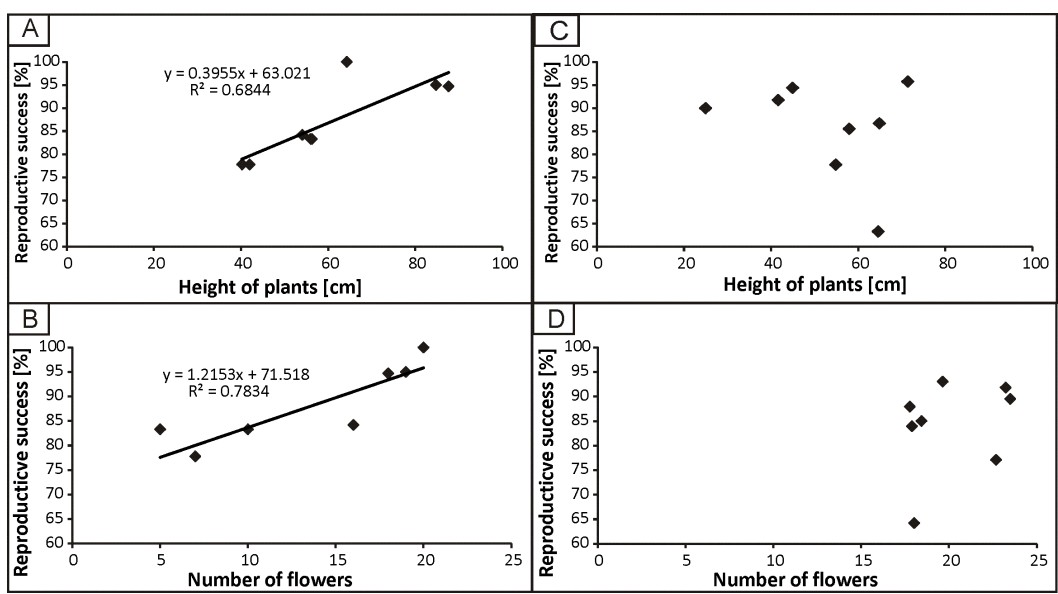

**Figure 5** The dependence of reproductive success from height of plants and number of flowers in populations from anthropogenic (A, B) and natural (C, D) habitats.

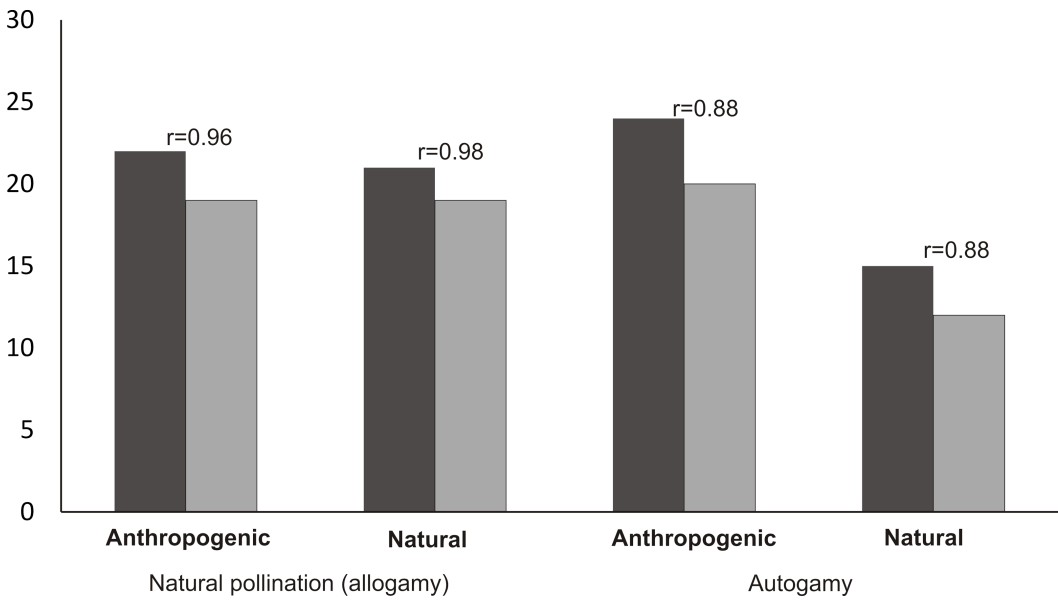

**Figure 6** Mean number of flowers and fruits in analyzed populations and Spearman correlation between number of flowers and fruits.

also strongly positively correlated (Spearman's correlation, respectively: $r = 96$ and $r = 98$, $p < 0.05$) with the number of flowers in the inflorescence.

In the population from the natural habitat, the number of fruits produced by open-pollination was slightly higher than the number of fruits produced by autogamy (Mann–Whitney $U$ test, $Z = 1.30$, $p = 0.48$). The proportion of dead seeds (with unstained embryo)

**Table 4  Ratio of dead and alive seeds developed in autogamy and natural pollination in analyzed populations.**

| Trait | Populations | | Mann–Whitney $U$ test, $p < 0.05$ |
|---|---|---|---|
| | Anthropogenic | Natural | |
| **Natural pollination (allogamy)** | | | |
| Live seed (%) | $49.7 \pm 0.5$ | $51.2 \pm 2.1$ | ns |
| Dead seeds (%) | $50.3 \pm 0.7$ | $48.8 \pm 2.6$ | ns |
| **Autogamy** | | | |
| Live seed (%) | $24.6 \pm 2.9$ | $29.5 \pm 3.7$ | $p < 0.05$ |
| Dead seeds (%) | $75.4 \pm 10.2$ | $70.5 \pm 8.6$ | $p < 0.05$ |

**Notes.**
ns, Non-significant results.

resulting from autogamy varied from 70.5% to 75.4%, and was higher compared to natural pollination (which varied from 48.8% to 50.3%) (Table 4).

## DISCUSSION

### Pollinators of *E. helleborine*

Pollinator availability is one of the key aspects of reproductive success of angiosperm species including orchids. According to *Kearns, Inouye & Waser (1998)* and *Potts et al. (2003)*, in approximately 90% of all angiosperm species, insect pollinators are involved in sexual reproduction of these plants. In orchids, approximately 70% of the species are closely related to specific insect pollinators (*Neiland & Wilcock, 1998*). Although our knowledge of how human-induced habitat disturbance affects diversity and composition of pollinator fauna is still fragmentary (*Aizen & Vázquez, 2006*); it is suggested that pollinator diversity decline is mainly due to the transformation of the environment associated with urbanisation, agricultural development and transformation of the area. Numerous studies have confirmed declining diversity and changes in species composition in cities and other anthropogenic habitats, especially in the case of bees (Apidae) and hoverflies (Syrphidae) (e.g.: *Schweiger et al., 2007*; *Banaszak-Cibicka & Zmihorski, 2012*). Regarding these two insect groups, we have found that only hoverflies occurred in much smaller numbers in anthropogenic habitats compared to natural ones, while in the case of bees, these values were very similar (Table 2). In contrast, the effect of small-scale disturbance can be positive for pollinator fauna, especially in forest habitats. *Quintero, Morales & Aizen (2010)* have noted that local species richness, abundance, and diversity of insect pollinators were higher in the case of disturbed areas than in natural ones. Our results clearly correspond with those latter observations, as in our studies in natural habitats members of only 14 insect families were noted, while in anthropogenic areas 19 families of insect pollinators occurred. Many authors suggest that higher insolation can provide an explanation of such a phenomenon (e.g., *Herrera, 1995*; *Thompson, 2000*; *Hegland & Boeke, 2006*). *Herrera (1995)* and *Quintero, Morales & Aizen (2010)* have noted that small patches and other forest gaps with high insolation can be characterised by higher pollinator activity, particularly of small-sized insects, which may increase local richness and abundance of day-active pollinators. Moreover, forest areas exposed to higher insolation are often

characterised by higher flower diversity and, as a result, by higher abundance of insect pollinators (*Thompson, 2000*; *Hegland & Boeke, 2006*). Sometimes it is even suggested that small-scale disturbed forest areas might act as "diversity oases" for flowering plants (*Romey et al., 2007*). The higher pollinator diversity in anthropogenic habitats of the studied sites may be explained by the higher insolation of anthropogenic habitats as the most common species were small and medium-size flowering plants, whereas the tree species dominated in natural habitats (Table S1). The anthropogenic habitats probably were located near the natural habitats, so the edge effects maybe also had an important role influencing the high pollinator diversity.

Traditionally, the orchid *E. helleborine* demonstrates different morphological and physiological adaptations to attract social wasps as pollinators (e.g., *Judd, 1971*; *Müller, 1988*; *Claessens & Kleynen, 2011*). According to the literature, its main pollinators are wasps belonging to the following genera: *Vespula*, *Vespa*, and *Dolichovespula* (*Claessens & Kleynen, 2011*). However, at least in some regions of the orchid distributional range, additional insect groups, such as flies and beetles, may play an important role in pollination (*Jakubska et al., 2005*; *Claessens & Kleynen, 2014*). As shown above, the pollinators of *E. helleborine* noted during our studies belonged to six orders and 24 insect families (Fig. 3). All these insects could be characterised by four different types of mouth-parts adapted to collect food in different ways (Fig. 4). Similar results were noted by *Jakubska et al. (2005)* who observed five coleopteran, four hymenopteran, two dipteran, and one lepidopteran family acting as pollinators of this orchid species and also belonging to four groups according to the type of mouth-parts. In comparison with our results, only piercing and sucking insects were not recorded as pollinators of *E. helleborine* by the above-mentioned authors, while, on the other hand, they noted sucking insects (Lepidoptera), which we did not observe. Such high taxonomical diversity of insects, as well as their diverse morphological adaptations of mouth-parts (all five main types of insect mouth-parts) used for collecting nectar and pollen, clearly suggest that *E. helleborine* is an opportunistic species according to pollinators. As it was shown by *Jacquemyn, Brys & Hutchings (2014)*, such a strategy can be also used in some other species belonging to this orchid genus. In their summary of knowledge concerning pollinators of *E. palustris*, the authors have noted members of six families of Coleoptera, 22 of Diptera, 12 of Hymenoptera, and one of Heteroptera (wrongly placed among Hymenoptera). However, it seems that a few insect groups play a much more important role in pollination biology of *E. helleborine* than the others. In our studies, both in natural and anthropogenic sites, sponging (flies, mainly Syrphidae) and chewing insects (mainly Vespidae and Formicidae but also Coleoptera) dominated, with the chewing-sucking insects (Apidae) as the third group according to the frequency of occurrence (Table 2, Figs. 3 and 4). Such differences may result from: geographical location, diversity of pollinator fauna, weather conditions, and especially air temperature, which may change emission of attractants contained in the nectar of *Epipactis* (*Ehlers & Olesen, 1997*). The important role of Vespidae, as well as of Syrphidae and Apidae, was observed not only in other Polish populations of *E. helleborine* (*Jakubska et al., 2005*), but also in other regions of Europe (e.g., *Claessens & Kleynen, 2014*). Similar patterns were also observed in other species of *Epipactis* such as *E. palustris* (*Vöth, 1988*; *Jacquemyn, Brys & Hutchings, 2014*),

*E. atrorubens* (*Jakubska-Busse & Kadej, 2011*), *E. consimilis* (*Ivri & Dafni, 1977*), *E. turcica* (*Fateryga, 2012*) and *E. veratrifolia* (*Jin et al., 2014*). As a result, at least some insect species visit *Epipactis* not only for its highly energetic pollen and/or nectar, but also to look for prey, i.e., other insects attracted by the flowers (*Rico-Gray & Oliveira, 2007*). For example, Vespidae, Crabronidae or Ichneumonidae, but also some Syrphidae flies, can be classified as such predators. As was shown by *Turlings, Tumlinson & Lewis (1990)* and *Brodmann et al. (2008)*, at least some *Epipactis* species (including *E. helleborine* and *E. purpurata*), produce green-leaf volatiles (GLVs) whose chemical composition is very similar to those emitted by damaged plant tissues when the plant is attacked by caterpillars. The latter are known as one of the most important prey for wasps. Surprisingly, *Jin et al. (2014)* have noted that females of some Syrphidae can lay their eggs on orchids attacked by aphids (Aphidoidea), which are the main food for their hatched larvae. The explanation of this phenomenon was provided by *Stökl et al. (2010)*, who noted that flowers of *E. veratrifolia* are visited by some aphidophagous Syrhipidae as the orchid produces $\alpha$- and $\beta$-pinene, $\beta$-myrcene and $\beta$-phellandrene. These substances are very similar to aphid-derived kairomones, which normally are emitted as alarm pheromones by several aphid species. Hoverflies were also noted as important pollinators of *E. helleborine*, both during our studies (Table 2, Fig. 3) and by *Jakubska et al. (2005)*. Moreover, aphids are regularly noted as feeding on this orchid species (A Rewicz, pers. obs., 2011–2013). Thus, we can suppose that such chemical mimicry is more common among *Epipactis* species than was shown until now. Overall, these results indicate that *E. helleborine* has a diverse group of pollinators, which may promote this species in very rapidly changing areas transformed by man and which is one of the key features of apophytes.

## Reproductive success and effect of autogamy of *E. helleborine* seeds

Autogamy in *E. helleborine* was observed by many authors, some of them (*Richards & Porter, 1982*; *Robatsch, 1983*) have claimed that this species shows optional autogamy (mixed-mating) (*Ehlers & Pedersen, 2000*; *Claessens & Kleynen, 2011*). However, *Ehlers, Olesen & Gren (2002)* have suggested that autogamy in *E. helleborine* is rare and that this phenomenon occurs only in specific conditions, i.e., when suitable pollinators are lacking. Our results provide evidence that autogamy occurs in populations from both anthropogenic and natural habitats of *E. helleborine* (Table 4). Despite some differences in the number of fruits between the populations of the two habitats, we have found no significant differences in the number of fruits formed between autogamy and natural pollination, which is congruent with work of *Weijer (1952)*. In our opinion, autogamy is a common phenomenon in the life cycle of *E. helleborine*.

According to *Grime*'s (*1979*) theory, some species can tolerate environmental disturbances. *Hágsater & Dumont (1996)* have suggested that orchids belong to the group between ruderal and stress-tolerant plants. Recent studies of *Rewicz, Kołodziejek & Jakubska-Busse (2016)* have highlighted a positive impact of disturbed anthropogenic habitats on occurrence of some orchids species, even against the general thesis that orchids are competitively weaker than other plant species. In this particular case, it may be caused by reduction in the vigour of other plants by some management practices. *Djordjević et al.*

*(2016a)* suggested that disturbed habitats can be preferred by orchids. It is possible that *E. helleborine* from the anthropogenic habitats has more space with a favourable light regime. Furthermore, some ecological conditions, such as soil moisture, soil pH, and organic matter could also be conducive to the growth of *E. helleborine*, which results in its larger size in anthropogenic habitats.

The relationship between plant height and reproductive success was confirmed by *Machaka-Houri et al. (2012)* in their studies on *Orchis galilea*, as well as on *Ferocactus cylindraceus*, *F. wislizeni*, and *Lotus corniculatus* (*Ollerton & Lack, 1998*; *McIntosh, 2002*). Our results also suggest that there is no association between reproductive success and the number of flowers on a sprout in populations from both anthropogenic and natural habitats. It appears that the height of the plant and the number of flowers in orchids enhances attractiveness for insect pollinators (*Kindlmann & Jersakova, 2005*). Specimens from the populations from the anthropogenic habitats we studied were higher and had more diverse pollinating fauna. Plants with bigger shoots and more flowers are more tempting for pollinators, which results in more efficient transport of pollinia (*Van der Piper & Waite, 1988*). Moreover, no significant correlation between density of plants and reproductive success in populations from the anthropogenic habitats was noted and in populations from the natural habitats such correlation was weak. Similar results were obtained in other studies (*Sih & Baltus, 1987*; *Ågren, 1989*; *Alexanderson & Agren, 1996*; *Ehlers, Olesen & Gren, 2002*).

In conclusion, our research demonstrates that the spectrum of insects pollinating *E. helleborine* is much wider than it has been suggested in the literature. Increased variety of possible pollinators allows faster and better adaptation to the human-changed environment. The reproductive success of *E. helleborine* was higher in anthropogenic habitats, which might have been a higher number of visits and greater species diversity of pollinators, as well as by a larger size of the plants. Moreover, autogamy was not uncommon as the reproductive strategy, and we found no significant differences between the number of fruits formed by autogamy and by natural pollination. In addition, this study contributes to a better understanding of why *E. helleborine* is one of the few Eurasian orchid species that has been naturalised in North America. The study confirms the general thesis that orchid species which are not highly specialised in relation to the type of pollinator have wider distribution ranges and are less rare than orchid species that have a high level of pollinator specialisation (*Swarts & Dixon, 2009*).

Summarising, our study helps to explain why *E. helleborine* is one of the (few) orchid species that manages to successfully use anthropogenic habitats in a manner comparable to that of natural ones. The question how wide the tolerance range of this species is still remains open. To answer the question, we plan further research on pollinator diversity and reproductive success of *E. helleborine* in other types of anthropogenic habitats (the surroundings of industrial facilities, fly ash, mine tailing, highway, railway, urban parks, etc.).

## ACKNOWLEDGEMENTS

The authors wish to thank Małgorzata Karczewska, Alicja Klejps, Damian Kanioski, Jakub Kierzkowski and Iwona Stefaniak for their assistance in the fieldwork. We also thank Michał Grabowski and Marta Koniarek for improving the final English version of the manuscript. We are also grateful to Vladan Djordjević and an anonymous reviewer for constructive comments on the manuscript. The second author would like to dedicate this paper to Barbara Szmalenberg.

### Funding

The authors received no funding for this work.

### Competing Interests

The authors declare there are no competing interests.

### Author Contributions

- Agnieszka Rewicz conceived and designed the experiments, performed the experiments, analyzed the data, contributed reagents/materials/analysis tools, wrote the paper, prepared figures and/or tables, reviewed drafts of the paper.
- Radomir Jaskuła and Tomasz Rewicz performed the experiments, analyzed the data, contributed reagents/materials/analysis tools, wrote the paper, prepared figures and/or tables, reviewed drafts of the paper.
- Grzegorz Tończyk contributed reagents/materials/analysis tools.

### Field Study Permissions

The following information was supplied relating to field study approvals (i.e., approving body and any reference numbers):

Regional Directoriate of Environmental Protection in Białystok (Regionalna Dyrekcja Ochrony Środowiska w Białymstoku) - permission no. WPN6400.74.2013.MW.

Ministry of the Environment (Ministerstwo Środowiska) - permission no. 35/17258/12/RS.

### Data Availability

The raw data has been supplied as Supplementary Files.

### Supplemental Information

Supplemental information for this article can be found online at http://dx.doi.org/10.7717/peerj.3159#supplemental-information.

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
