# Peer review of "Pollinator diversity and reproductive success of Epipactis helleborine (L.) Crantz (Orchidaceae) in anthropogenic and natural habitats"

_PeerJ, doi:10.7717/peerj.3159_

## Round 0.1 · original submission · Major Revisions

Please revise, paying careful attention to the reviewers' comments. Be advised that the revised version will likely go out for review again.

·

Basic reporting

No Comments

Experimental design

No Comments

Validity of the findings

No Comments

Additional comments

Manuscript #10923

Pollination biology and reproductive success of Epipactis helleborine (L.) Crantz (Orchidaceae) in anthropogenic and natural habitats

Reviewer comments:

The authors investigated the reproductive success, pollinator diversity and autogamy of the orchid species Epipactis helleborine in natural and anthropogenic habitats. They showed that the reproductive success among populations in anthropogenic habitats is significantly higher than among populations in natural habitats and that autogamy is not uncommon in the reproductive strategy of this species. Beyond this general purpose, they also provided new information about the pollination ecology of this orchid species. Overall, this paper highlights the importance of anthropogenic habitats in the survival of E. helleborine. Performed statistical analyses and interpretation of the results are globally and scientifically relevant. In order to improve the manuscript, specific comments and suggestions are described below (line numbers are aligned with the PDF document and the Word document from the authors):


Title
I suggest that the authors use the term "pollinator diversity" instead of "pollination biology".

Abstract

PDF: Page 5, Line (17-19); Word document: Page 2, Line (18-20)
Epipactis helleborine does not inhabit only the shaded forests, but also the semi-open forest communities, forest edges and clearings in open or dense woodland. So, I suggest to delete the word "shaded". Furthermore, it is better to write "town parks and gardens" instead of "towns, parks and gardens". Correct the sentence as follows:

"Epipactis helleborine is an Eurasian species which prefers woodland environments but it may also spontaneously and successfully colonise human-made artificial and disturbed habitats such as roadsides, town parks and gardens."

PDF: Page 5, Line (21, 23); Word document: Page 2, Line (22, 24)
I suggest that the authors use the term "pollinator diversity" instead of "pollination biology".

PDF: Page 5, Line (22); Word document: Page 2, Line (23)
Write "helleborine" instead of "helloborine".

PDF: Page 5, Line (36, 37); Word document: Page 2, Line (38, 39)
Rewrite this sentence as follows: "We suggest that higher reproductive success of E. helleborine in populations in anthropogenic habitats than in populations in natural habitats may result from higher number of visits by pollinators and their greater species diversity but also from the bigger size of plants growing in such habitats."

Introduction

PDF: Page 6, Line (45-47); Word document: Page 3, Line (47-49)
Please, correct the sentence as follows: "Orchidaceae is one of the most diverse and species-rich (20 000 - 30 000 species) plant families (Bauman, Kunkele & Lorenz, 2010; Djordjević et al., 2016) with many species that are seriously endangered and require conservation efforts to maintain their populations."

Correct the number of the year in the reference Djordjević et al., 2016.

PDF: Page 6, Line (47, 48); Word document: Page 3, Line (49,50)
Rewrite this sentence as follows: On the other hand, numerous orchid species grow in anthropogenic habitats. Please, reduce the number of references here.

PDF: Page 6, Line (52); Word document: Page 3, Line (54)
Authors should indicate which orchid species also grow in anthropogenic habitats. Add few sentences discussing the results of the new studies and indicate which orchid species were found to grow in anthropogenic habitats.

For example, the recent study of orchids in Turkey showed that graveyards are places where orchid species occur frequently (Löki & al. 2015). In addition, Djordjević et al. (2016) noted that Himantoglossum calcaratum, Anacamptis pyramidalis and Ophrys species often grow in habitats along the roads. Moreover, the same authors showed that Orchis purpurea was determined as an indicator of ruderal habitat type. Jurkiewicz et al. (2001) noted that populations of Epipactis atrorubens, E. helleborine, and Dactylorhiza majalis were observed on several mine tailings in southern Poland.

Löki V., Tökölyi J., Süveges K., Lovas–Kiss A., Hürkan K., Gábor S. & Molnár A. V. 2015. The orchid flora of Turkish graveyards: a comprehensive field survey. Willdenowia 45: 231–243.

Djordjević V., Tsiftsis S., Lakušić D., Jovanović S. & Stevanović V. 2016. Factors affecting the distribution and abundance of orchids in grasslands and herbaceous wetlands. Systematics and Biodiversity 14(4): 355–370.

Jurkiewicz A, Turnau K, Mesjasz-Przybylowicz J, Przybylowicz W, Godzik B. 2001. Heavy metal localisation in mycorrhizas of Epipactis atrorubens (Hoffm.) Besser (Orchidaceae) from zink mine tailings. Protoplasma 218: 117-124.

PDF: Page 6, Line (53); Word document: Page 3, Line (55)
Provide the full Latin name of the species with the authors: Epipactis helleborine (L.) Crantz

After the first time stating the full name of the species with the authors, later in the text, use only the abbreviation: E. helleborine.

PDF: Page 6, Line (53-55); Word document: Page 3, Line (55-57)
I suggest to remove old references and to use the new (recent) ones in the sentence about the distribution of Epipactis helleborine.

Kolanowska, M. (2013). Niche Conservatism and the Future Potential Range of Epipactis helleborine (Orchidaceae). PLoS ONE 8(10): e77352. doi:10.1371/journal.pone.0077352

Tranchida-Lombardo, V., Cafasso, D., Cristaudo. A., Cozzolino, S. (2011). Phylogeographic patterns, genetic affinities and morphological differentiation between Epipactis helleborine and related lineages in a Mediterranean glacial refugium. Ann. Bot. 107, 427–436.

PDF: Page 6, Line (55, 56); Word document: Page 3, Line (57, 58)
I suggest not to write about the habitat preferences and the colors of the flowers in the same sentence. Please, rewrite the sentences and reduce the number of references and sentences.

Authors should write that E. helleborine grows usually in deciduous and coniferous forest communities, edges and clearings in woodland, up to 2 000 m a.s.l. (Delforge, 2006). Furthermore, this species inhabits different types of anthropogenic habitats (roadsides, cemeteries, poplar plantations, gravel pits, quarries, railway embankments, mine tailings) and may also appear spontaneously in urban areas such as town parks and gardens (Kolanowska, 2013). Some studies showed that E. helleborine is a species with broad ecological tolerance which is not highly specialized (Tsiftsis et al., 2008).

Kolanowska, M. (2013). Niche Conservatism and the Future Potential Range of Epipactis helleborine (Orchidaceae). PLoS ONE 8(10): e77352. doi:10.1371/journal.pone.0077352

Tsiftsis, S., Tsiripidis, I., Karagiannakidou, V., & Alifragis, D. (2008). Niche analysis and conservation of the orchids of east Macedonia (NE Greece). Acta Oecologica, 33, 27–35.

PDF: Page 7, Line (62, 63); Word document: Page 3, Line (64, 65)
Please, reduce the number of references here. You can use one or two most important ones.

PDF: Page 7, Line (63-65); Word document: Page 4, Line (66-68)
Write the reference at the end of the sentence.

Orchids that colonize anthropogenic habitats are characterized by a specific set of features: fast growth resulting in large plant size, fast flowers production and light anemochoric seeds (Forman et al., 2009).

PDF: Page 7; Word document: Page 4
Replace and rewrite the sentence "Epipactis helleborine may produce seeds in allogamy and optionally, in autogamy (Tałałaj&Brzosko 2008)." after the sentence: "Epipactis helleborine may produce from 1000 to 2200 seeds in one fruit (bag) (Arditti & Ghani, 2000; Rewicz, Kołodziejek & Jakubska-Busse, 2016)."

PDF: Page 7, Line (70, 71); Word document: Page 4, Line (72-73)
Please, reduce the number of references here and use one (optional) or two.

PDF: Page 7, Line (74-77); Word document: Page 4, Line (77-79)
Please, use more recent or more relevant literature sources and rewrite the sentences as follows: "However, the high number of seeds does not lead to high recruitment of seedlings which can be the result of specific biology of orchids, i.e. obligatory presence of mycorrhizal symbionts during germination and further plant growth."

For e.g., Rasmussen, H. (1995). Terrestrial orchids from seed to mycotrophic plant. Cambridge: Cambridge University Press.

PDF: Page 7, Line (77-79); Word document: Page 4, Line (79-82)
Rewrite the sentence as follows: "Low reproduction success, which is defined as the ratio of the number of flowers to the number of fruits (Doust & Doust, 1988), may also arise from high level of morphological adaptation of flowers to particular pollinators."

PDF: Page 7, Line (79-81); Word document: Page 4, Line (82-84)
Please, omit the sentence: "Some orchid species developed specific mechanisms such as: flower traps in Cypripedium calceolus or sexual deception and pseudo-copulation in Ophrys spp."

Instead of this sentence, provide some information about the differences in reproductive success between deceptive and rewarding orchids. For example, the studies showed that reproductive success in deceptive orchids is lower than that in rewarding ones (Kindlmann & Jersakova, 2006).

Kindlmann, P., Jersakova, J. (2006). Effect of Floral Display on Reproductive Success in Terrestrial Orchids. Folia Geobotanica 41: 47-60.

PDF: Page 7, Line (81, 82); Word document: Page 4, Line (84)
Start a new line with the sentence: "Human disturbance as habitat transformation is regarded as a principal cause of pollinator decline in global scale (Goulson et al., 2008)."

PDF: Page 8, Line (87-90); Word document: Page 5, Line (91-93)
Authors should mention the previous study (Ehlers et al., 2002) about the reproductive success of E. helleborine here. At the same time, they should indicate what is investigated, and about which little is known, i.e. have not been investigated. For example, Ehlers et al. (2002) investigated the reproductive success of E. helleborine in natural habitats. There is a lack of knowledge concerning the reproductive success and pollinator diversity of E. helleborine in relation to a certain type of natural and anthropogenic habitats.

Ehlers BK, Olesen JM, Gren JA. (2002). Floral morphology and reproductive success in the orchid Epipactis helleborine: regional and local across-habitat variation. Plant Systematic and Evolution 236: 19–32.

PDF: Page 8, Line (91,92); Word document: Page 5, Line (94-96)
Rewrite this sentence as follows: "In this study we compared the reproductive success and pollinator diversity of E. helleborine from natural and anthropogenic habitats."

Materials and methods

Reproductive success

PDF: Page 8, Line (103); Word document: Page 5, Line (106)
Authors should indicate that the study area is located in Poland, and to indicate the calendar period when the research was conducted. An illustrative and positive example is in the section Flower visitors.

PDF: Page 8, Line (106); Word document: Page 5, Line (110)
Indicate the species name of the pine species in the text and in the image caption (Figure 1). Rewrite as "Pinus sylvestris forest".

PDF: Page 9, Line (108, 109); Word document: Page 5, Line (111, 112)
I suggest to write the full name of the plant community "Galio sylvatici-Carpinetum betuli Oberd. 1957", and not to use the abbreviation "Galio-Carpinetum Oberd. 1957".
Did the authors use phytosociological methods to determine the existence of this plant community or the data were taken from previous research?

PDF: Page 8, Line (104-107); Word document: Page 5, Line (107-110)
Anthropogenic habitats can also be defined at the level of plant communities, as the authors did when it comes to natural habitats. Do the different habitat types described here may correspond to other detailed classification of ecosystems like EUNIS? Also in image caption (Figure 1).

PDF: Page 9, Line (110); Word document: Page 5, Line (114)
I suggest to add information of altitude for each studied site in the Table 1.

Authors should correct the terms in the Table 1. Population size is the number of individual organisms in a population, and not the size of the area occupied by the population in square meters. Therefore, instead of "Population size", authors should write "Area (m2)". At the same time, in the Table 1 "Number of shoots" corresponds to the term "Population size (Number of shoots)".

In addition, correct the number in the Table 1: density in the A1 population.
127 / 36 = 3.5277, so the density is 3.53.

PDF: Page 9, Line (114); Word document: Page 6, Line (117)
Write shoots/m2 instead of shoots/m2

Methods of sampling should be clarified. Authors should indicate how they determined the population size i.e. the number of shoots (probably counting the total number of individuals) and the size of the sampling area on which the number of individuals was determined. Why didn't the authors use a particular standard sample size? Please indicate whether the area of a square or a rectangle was used.

Flower visitors

Rewrite the title as "Pollinators" or "Pollinators of E. helleborine" instead of "Flower visitors". Authors used the subtitle "Pollinators" in the section Discussion.

I suggest not to write "anthropogenic populations" of E. helleborine and one "natural population". Correct and write "parking" instead of "parkin". Rewrite as follows: "Pollinators were both caught and observed in two populations of E. helleborine in two anthropogenic habitats: roadside in Guszczewin avillage (A1), and close to the car parking Hajnówka city (A2), and in one population in a natural habitat in the Białowieża Primeval Forest (N3)."

I suggest to omit this sentence because the authors did not use temperature as a variable in the analyzes in this study: "Air temperature was measured every two hours with Volkraft thermometer." In addition, the authors did not present temperature values in this study.

Provide some literature sources that were used in the identification of insect taxa or references that were used for the nomenclature.

PDF: Page 9, Line (126); Word document: Page 6, Line (129)
Write "helleborine" instead of "helloborine".

Data analysis

PDF: Page 10, Line (145); Word document: Page 7, Line (149)
Use italic U and rewrite as follows: Mann-Whitney U test. Also in the section Results (Autogamy) and Table 4.

Results

Flower visitors

PDF: Page 10, Line (152); Word document: Page 7, Line (157)
I suggest to rewrite the title as "Pollinators" or "Pollinators of E. helleborine".
I suggest to rewrite the title of Table 2 as follows: Pollinators of Epipactis helleborine in natural and anthropogenic habitats.

The authors presented the Fig. 3, but did not refer to it in the text. For example, add Fig. 3 in the sentence: "Pollinators of Epipactis helleborine collected during this study belonged to six orders and 23 families of insects (Tab. 2, Fig. 3)."

In the Table 2, write "Habitat type" instead of "Site type".

PDF: Page 11, Line (154-157); Word document: Page 7, Line (159-162)
I suggest not to write "anthropogenic populations". Rewrite the sentence as follows: "In case of the populations of anthropogenic habitats, the taxonomic diversity of pollinators was higher, with 19 families grouped in five orders, while in the natural habitat, we noted only 14 families from four orders (statistically significant values, chi-squared test, p = 0.00)."

PDF: Page 11, Line (159,160); Word document: Page 8, Line (166)
Rewrite as follows: "Coleoptera were the third main group of pollinators making up 6% in the populations of anthropogenic habitats and 4% in the population of natural habitat."

PDF: Page 11-12, Line (163, 182); Word document: Page 8-9, Line (169, 189-190)
I suggest to rewrite as follows: "the populations of anthropogenic habitats" instead of "anthropogenic populations"

PDF: Page 11, Line (169, 170); Word document: Page 8, Line (176)
I suggest to rewrite as follows: "In the population of natural habitat" instead of "In the Białowieża Primeval Forest".

Autogamy
PDF: Page 12, Line (187); Word document: Page 9, Line (194)
I suggest to rewrite as "the populations of anthropogenic habitats" instead of "anthropogenic populations"

PDF: Page 12, Line (189-190, 191-192); Word document: Page 9, (196-197, 199)
I suggest to rewrite as follows: "Both in the populations in natural and anthropogenic habitats" instead of "Both in the natural and anthropogenic populations".

Reproductive success

PDF: Page 13, Line (200-209); Word document: Page 9, Line (207-217)
I suggest that the authors use "populations in anthropogenic habitats" instead of "anthropogenic populations". In addition, use "populations in natural habitats" instead of "natural populations".

Fig. 6
I suggest to rewrite as follows: "The dependence of reproductive success in populations in anthropogenic habitats: a) the height of plants, b) the number of flowers."

Discussion

Pollinators

PDF: Page 13, Line (214-217); Word document: Page 10, Line (223-225)
Please, reduce the number of references here (use one to three references).

PDF: Page 14, Line (222); Word document: Page 10, Line (230, 231)
Please, reduce the number of references here (for example, use Jakubska et al., 2005a, b; Claessens & Kleynen 2014).

PDF: Page 14, Line (232); Word document: Page 11, Line (241)
Write "helleborine" instead of "helloborine".

PDF: Page 14, Line (242); Word document: Page 11, Line (252)
What do you mean by "taxonomy of pollinators fauna" in the sentence below?

Such differences may result from: geographical localization, taxonomy of pollinators fauna, weather conditions and especially air temperature, which may change emission of attractants contained in nectar of Epipactis (Ehlers & Olesen, 1997).

PDF: Page 15, Line (247-251); Word document: Page 11, Line (256-260)
Please, reduce the number of references in the sentence. Better to use some common references that apply to all causes or the majority of listed species.

It is also known for another species of Epipactis like: E. palustris (eg. Nilsson, 1978; Brantjes, 1981; Verbeke & Verschueren 1984; Vöth, 1988; Jakubska-Busse & Kadej, 2008; Fateryga, 2012; Jacquemyn, Brys & Hutchings, 2014), E. atrorubens (Jakubska-Busse & Kadej, 2011), E. consimilis (Ivri & Dafni, 1977), E. turcica (Fateryga, 2012) and E. veratrifolia (Jin et al., 2014).

PDF: Page 16, Line (270, 271); Word document: Page 12, Line (280, 281)
Authors used a similar sentence with the same reference in the section Introduction. Omit the sentence: "This is all the more urgent problem because about 70% of species is closely related to the specific pollinator species (Neiland & Wilcock, 1998)" or omit the similar sentence in the Introduction.

In the Introduction, the authors wrote: "Orchids are even more prone to that adverse trend, because up to 70% of species are pollinated by particular species of pollinator (Neiland & Wilcock, 1998)."

Reproductive success and effect of autogamy of E. helleborine seeds

PDF: Page 16, Line (279-281); Word document: Page 13, Line (289-292)
Reduce the number of references and use the recent ones or references that include other older references.

PDF: Page 16, Line (283, 284); Word document: Page 13, Line (294)
Omit the comma in the sentence: "Our results proved,.." and rewrite as follows: "Our results proved that..."

PDF: Page 17-18; Word document: Page 13-14
The authors should explain how the species E. helleborine in anthropogenic habitats has a larger size of plants and higher diversity of pollinators. Authors should add some sentences that explain why orchids grow in anthropogenic habitats. At the same time, this will in some way, explain the higher reproductive success of this species in anthropogenic habitats.

For example: Hágsater and Dumont (1996) noted that orchids belong to the group between ruderal and stress-tolerant plants according to Grime's (1979) theory, suggesting that they can tolerate some degree of disturbance. Since the orchid species are generally competitively weak species, a certain degree of disturbance can positively affect orchid performances, because of the reduction in the vigour of competing species by some management practices. In addition, Djordjević et al. (2016) highlighted the ecotone characteristics of the preferred habitats of some orchid species and noted that they can tolerate a certain degree of disturbance. It is possible that E. helleborine in anthropogenic habitats has more space with a favorable light regime. In addition, the edge effects may explain the higher diversity of insects in anthropogenic habitats. Furthermore, some ecological conditions, such as soil moisture, soil pH, organic matter, could also be the reason why individuals of E. helleborine have a larger dimension in anthropogenic habitats.

Hágsater, E., Dumont, V. (Eds.) 1996. Orchids: Status, Survey and Conservation Action Plan. IUCN, Gland, Switzerland and Cambridge, UK.

Grime, J.P. 1979. Plant strategies and vegetation processes. John Wiley & Son, Chichester.

Djordjević V., Tsiftsis S., Lakušić D., Jovanović S. & Stevanović V. 2016. Factors affecting the distribution and abundance of orchids in grasslands and herbaceous wetlands. Systematics and Biodiversity 14(4): 355–370.

PDF: Page 17, Line (300); Word document: Page 13, Line (310)
The authors wrote that specimens from anthropogenic populations had "bigger flowers". Is this just an observation? If the authors measured the size of the flowers, why the results are not presented in the Results section?

PDF: Page 17, Line (304); Word document: Page 14, Line (315)
Authors wrote that their results confirm the theses that simultaneous opening of flowers in anthropogenic populations increases the number of pollinator visits and pollinia transport. Did the authors investigate the number of pollinator visits? In my opinion, better to omit the sentence: "Our results confirm these theses."

Conclusion
Authors should add some sentences suggesting that this study confirms the statement that orchid species that are not highly specialized in relation to the type of pollinator have wider distribution and are not as rare as compared to those orchid species that have a high level of specialization (Swarts & Dixon, 2009). In addition, this study contributes to a better understanding of why E. helleborine is one of the few Eurasian orchid species that have been naturalized in North America.

Swarts, D. N., & Dixon, W. D. (2009). Terrestrial orchid conservation in the age of extinction. Annals of Botany, 104, 543–556. doi: 10.1093/aob/mcp025

References
Authors should reduce the number of old and local references and use the recent and the most important ones.

Add the volume number and page numbers in the reference, and correct the year number:

Djordjević V, Tsiftsis S, Lakušić D, Stevanović V. 2016. Niche analysis of orchids of serpentine and non-serpentine areas: Implications for conservation. Plant Biosystems 150(4): 710-719 doi: 10.1080/11263504.2014.990534

Reviewer 2 ·

Basic reporting

Mostly, the article is written in English using clear and unambiguous text. However there are parts should be improved for better understanding. I provide some, however incomplete, suggestions for improvement below (General Comments – More specific comments).

I think the article does not include sufficient introduction and background to demonstrate how the work fits into the broader field of knowledge. On the one hand, the main focus of the paper, the comparison of anthropogenic and natural populations, is insufficiently introduced. On the other hand, a lot of information that is not essential for the main focus is provided. In a few cases not the relevant literature is cited. For more detailed comments, please see below (General Comments).

In general Figures are relevant to the content of the article. However, I think Fig. 2 is redundant, because the text describes the allogamy experiment well enough. In general, the labelling of the figures is fine. Only in Fig. 6 it is not clear what kind of data is shown. Please indicate this in the figure legend or in the main text. Fig. 3 is not cited in the main text.

The authors do not provide the raw data. They only include the processed data (means). When only providing means, a lot of information is lost. Thus, I think, for better understanding the raw data should be provided.

Experimental design

The article contains defined research questions. However, there is no clear hypothesis driving these questions and it is not clearly stated how the research questions help to better understand how orchids/plants deal with human altered habitats (the main focus of the article). Focusing the introduction more on the central topic and providing a hypothesis would greatly improve the whole article. For more detailed comments, please see below (General Comments).

In some parts, the methods are not described with sufficient information. Please see below (General Comments), for which information I think is missing.

Validity of the findings

The data are not always statistically sound or not enough information is given to judge it. Please see below (General Comments), for what my concerns are and what should be changed or which information should be added.

Not all data on which the conclusions are based are provided or made available. I think the raw data should be provided. See also my comment above (Basic Reporting).

In the conclusions, I miss a clear connection to the original research question. As in the introduction, focussing more on the central topic would improve the article. Sometimes the discussion is based on findings that are not reported in the results section. For more detailed comments, please see below (General Comments).

There is hardly any speculation. However, I think it would be really interesting to read the authors’ thoughts about potential implications of their results.

Additional comments

In general, I think this is a nice study with a simple design, reporting a higher reproductive success of E. helleborine in anthropogenic populations, a considerable extent of autogamy, which did not differ between natural and anthropogenic populations, and a more diverse pollinator community in anthropogenic populations. The results are potentially interesting and may further our understanding of how human disturbed habitats affect orchids (and maybe other plant species as well). However, I have some concerns regarding the distinctiveness of natural and anthropogenic populations, the statistical analyses, and the interpretation of the data. Moreover, I miss a clear focal point in the article. Below, I list my concerns more detailed.



General comments

Abstract

In my opinion, it is (a bit) exaggerated to talk about a ‘much’ wider spectrum of pollinating insects (line 40 and see also line 233 in the discussion) because the literature cited already indicates that E. helleborine is pollinated by a broad range of insects.


Introduction

Reading the introduction, leaves me uncertain of what is the main focus of the article. According to the title and the first paragraph of the introduction, the focus is the difference between natural and anthropogenic populations of E. helleborine. There is a lot of information about the biology of E. helleborine and orchid reproduction, which of course is interesting, but which distracts the reader from grasping the actual topic of the article. In lines 53-67, there is a lot of information about the biology of E. helleborine, which is not relevant for the introduction and should be moved to the ‘materials and method’ section and described there in a paragraph on the study species. In lines 68-90, I do not see how dust seeds, the orchids’ dependency on mycorrhizal symbionts for germination and growth, and the specific pollination mechanism of some orchids is related to the topic of differences between natural and anthropogenic populations of E. helleborine. As these aspects are anyway not consider in the study, deleting it would improve the readability and understandability of the introduction.

Moreover, the introduction lacks a hypothesis leading to the study questions, which would help the reader to understand why the study was conducted and why it was conducted the way it was. In particular the expression ‘… we wanted to find the differences (if any) in …’ (line 91) gives me this impression.


Materials and methods

In the materials and methods section, several points (listed below) are not unambiguously or detailed enough explained or are not mentioned at all.

In line 101+102, it is written ‘the ratio of the number of flowers to the number of seeds set (fruits)’ was used as a measure of reproductive success. Were the seeds or the fruits counted? In orchids, the number of seeds is certainly much higher than the number of fruits. From information later in the article, I assume that fruit set and not seed set was considered. Please, unambiguously describe what was the measure of reproductive success.

From the description and the photographs in Fig. 1, I am not convinced that E. helleborine anthropogenic populations represent anthropogenic habitats very well. I do not think they look very different from the natural populations. As I understand, E. helleborine, in general, does not grow in the thick understorey but always needs some kind of opening. The only difference I can tell from the pictures is that humans created these openings but otherwise their influence seems to be minor. In my opinion, E. helleborine populations in towns, parks, and gardens where they also occur (e.g. line 58) would be better examples. Please elaborate on how the two habitat types exactly differ, i.e. what is the exact extend of human influence in the anthropogenic populations. Does the soil, the surrounding vegetation etc. differ between the two habitat types?

In Tab. 1, there is s column labelled ‘population size (m2)’. Please describe how population size was estimated.

In lines 113+114, it is written that population density was measured. As the importance and potential differences of population density is not introduced in the introduction, it is not clear to me why it was measured. Please elaborate on this.

How many hours were spent catching and observing pollinators in each population? Is the number of hours comparable between anthropogenic and natural populations?

What kind of material was collected (lines 127+128)?

In lines 133+134, suddenly mouth-part types of pollinators are mentioned. As this is not mentioned in the introduction, I do not really understand its significance for this study. Please elaborate on what the significance of grouping pollinators according to their mouth-part types is in regard to pollination and potential differences between anthropogenic and natural populations in the introduction.

What and how many control plants were used in the autogamy experiment (lines 136-141)?

What were the sample sizes for the statistical analyses (lines 144-149)? Please describe this here and also mention the sample sizes or the degree of freedom, respectively, in the results section whenever giving the statistics. As I understand, means for populations and years were used. If data for each individual plant were used for the statistical analyses, individuals from the same population and year would not be independent data points. In general, I think sample sizes are very low when means were used for statistical analyses. Like this, a lot of information and statistical power is lost. The results might be much stronger, if data points for all individuals were included and mixed models controlling for population and year effects were conducted.


Results

For consistency and better understanding, please use the same order of the subtitles reproductive success, flower visitors, and autogamy in the materials and methods section as well as in the results section. Alternatively, make clear why a different sequence was used.

In my opinion, the results are at several places (listed below) not described detailed enough, contradictory or the statistics are missing. The statistics should also be given in cases where it is not significant.

To me, it is not clear whether all the differences in pollinator communities between anthropogenic and natural populations described in lines 163-185 are real or just by chance difference. Firstly, in the materials and methods section it is written that a Chi-squared test was used to assess differences in the diversity of pollinator communities between the two habitat types. However, I cannot find the results of this statistical analysis. Thus, I wonder whether all the differences described in the results section are actually statistical meaningful. Please add the statistics. Secondly, two anthropogenic populations but only one natural population were investigated. As the results are not shown separately for the two anthropogenic populations, I am not convinced that the pollinator community in anthropogenic populations are really more variable. If the pollinator communities differ considerably between anthropogenic populations, it could well be that the pollinator communities in each anthropogenic population is less diverse compared to the natural population. Please elaborate on this.

In the materials and methods section it is written that reproductive success was measured as the ratio between the number of fruits and the number of flowers on the inflorescence. Thus, I do not see why suddenly the number of fruits was used as a measure of reproductive success when analysing differences in autogamy between the two habitat types (lines 187-191). Please elaborate on this. Moreover, did the number of flowers per inflorescence differ?

In lines 194-196, it is written that the difference between autogamy and natural population was statistically non-significant but the p value shown is p < 0.05. What is correct?

In lines 201-203, ranges between populations are described, whereas in lines 203+204 ranges between years are described. Please be consistent or describe why not the same results are given in both cases.

In lines 205-209, it is written that correlations were used, whereas in lines 203+204 it is written that regression analyses were done (which are not mentioned in the paragraph on the data analysis in the materials and methods section). Were correlation or regression analysis used? The former seems to be more appropriate as sample size is very low. Please describe which statistical analyses were done in the paragraph on data analysis.

For a better understanding, also the graphs of the natural populations should be included in Fig. 6 (lines 211). Moreover, why are there 7 data points in Fig. 6? It should be only 4 (4 anthropogenic populations) or 8 if a separate data point was used for each year (which I do not think is appropriate because data from the same population are not independent). Please describe which data are shown in Fig. 6.


Discussion

In general, I think, the discussion is held very general and not focused on the main topic of the article: anthropogenic versus natural populations (at least I think that should be the main focus according to the title). Moreover, the discussion often only repeats the results instead of interpreting them and lacks potential implications for E. helleborine growing anthropogenic versus in natural populations. In addition, I think that sometimes the authors claim something they cannot really prove. Below I elaborate on these points.

For example, more than two pages are used to discuss pollination in E. helleborine in general, and there is only a very short and unspecific paragraph on the potential meaning of the results in relation to anthropogenic versus natural populations. I think the readability and understandability of the discussion would improve a lot if the former part was deleted or strongly shortened and the latter part was extended and elaborated on. Moreover, I think, it is (a bit) exaggerated to talk about E. helleborine being a ‘much’ more opportunistic species according to pollination (line 233 and see also line 40 in the abstract and line 309 in the conclusions) because the literature cited already indicates that E. helleborine is pollinated by a broad range of insects. Thus, this neither warrants spending two pages on the general pollination biology of E. helleborine.

In lines 291+292, it is written that the number of visits was higher (see also line 312). However, as I understand, the number of visits was not investigated (even though it would have been a really interesting measure) and cannot be used as an argument. If the number of visits was counted, please describe that in the materials and methods section.

There are many more studies that show a relationship between plant height and reproductive success (lines 294+295) and should be cited here.

In lines 299+300, it is written that individuals from anthropogenic populations had ‘bigger flowers’. However, as far as I understand from the article, flower size was not measured at all. If it was, please specify that in the materials and methods section. Or is inflorescence size meant? This measure is listed in line 113 but I cannot find the data for this measure. Please include it in Tab. 5 and provide the statistics in the results section.

In line 304, it is written that the results confirm the hypothesis of MacConaill (1998). However, I do not see how they could because the present study did not investigate how simultaneous flowering was in anthropogenic and natural populations. Please elaborate on this.

Please elaborate on the implication of the findings of the relationship between reproductive success and density in anthropogenic and natural populations (lines304-307).

In the conclusions (lines 308-314), some potential implications of the results found in this study are listed. However, they are not discussed and it is not shown how the data of the study support this implications. These aspects should be discussed earlier in the discussion. In addition, I miss, what the significance of the findings of this study is.



More specific comments

line 17: I think it would be clearer, if it was written more specifically; e.g. ‘orchid species’ instead of only ‘species’.

line 22: I think it would be clearer, if it was written more specifically; e.g. ‘between natural and anthropogenic habitat types’ instead of ‘from different habitat types’.

line 23+24: ‘were compared’ should be before ‘in populations of …’.

line 25: ‘were studied’ should be before ‘in two seasons’.

lines 29+30: Seed set was not considered in the study. Thus, simplify the information in brackets as follows: ‘the ration of the number of flowers to the number of fruits’.

line 35: ‘respectively’ should be at the end of the sentence.

line 47: Replace ‘On the other side’ by ‘However’ (if there is no ‘on the one side’ there cannot be an ‘on the other side’ either).

line 68: ‘A reproductive system’ is probably essential for the population biology of any species. Should it be ‘the unique reproductive system’ instead of ‘a reproductive system’?

line 70: It is called ‘dust seeds’ and nod ‘seed dust’.

lines 70+71: The cited literature is not appropriate here. On the one hand, articles are not related to the topic (e.g. Rasmussen & Whigham, 2002) and, on the other hand, there is no literature referring to the expression of ‘dust seeds’.

line 72: What is meant with ‘bag’?

line 78: Seeds or fruits? Is the difference relevant?

line 83: Replace ‘more’ by ‘particularly’ and ‘species’ by ‘all species’.

lines 85+86: Replace ‘seeds production’ by ‘seed production’.

line 93: ‘the pollinator fauna’ instead of ‘pollinator fauna’.

line 97: Replace ‘populations’ by ‘habitats’.

line 101: ‘the definition’ instead of ‘definition’.

lines 115-117: Please, embed this as a sentence in the text.

line 122: are the other five populations not located in eastern Poland?

line 123: ‘from the fresh orchid flowers’ should be before ‘using entomological hand nets’.

lines 143-149: For better understanding, please describe the statistical analyses in the same order as the investigations are mentioned in the materials and methods section above (i.e. reproductive success – flower visitors – autogamy).

line 157: ‘p = 0.00’ should be ‘p < 0.01’.

line 159: Fig. 5 does not show these results and thus should not be cited here.

line 189: Is it p =0.0008 or p < 0.05? I think the former is the test statistics and should not be ‘p’.

lines 191-193: Please add the statistics here.

line 194: It should be ‘produced by’ instead of ‘produced in by’.

line 196: Is it p =0.48 or p < 0.05? I think the former is the test statistics and should not be ‘p’.

line 196: It should be ‘The amount’ instead of ‘amount’.

line 197: ‘It should be compared’ instead of ‘comparing’.

line 201: Is it p =0.0008 or p < 0.05? I think the former is the test statistics and should not be ‘p’.

line 201: Fig. 6 does not show these results and thus should not be cited here.

line 204: It should be ‘from 15 (2011) to 14 (2012)’ instead of ‘from 15 (2011) do 14 (2012)’.

line 220: The expression ‘on the other hand’ should only be used when the expression ‘on the one hand’ is used above.

lines 223+224: Here and in Tab. 2 it is written 24 insect families but in the results (line 154) it is written 23 insect families. What is correct?

line 323: It should be ‘E. helleborine is a’ instead of ‘E. helleborine is’.

line 237: The expression ‘on the other hand’ should only be used when the expression ‘on the one hand’ is used above.

line 238: it should be ‘play a much more’ instead of ‘play much more’

lines 238+239: It should be ‘than the others’ instead of ‘that the others’.

line 252: It should be ‘by the flowers’ instead of ‘by flower’.

line 260: It should be ‘are the main’ instead of ‘are main’.

line 290: The expression ‘on the other hand’ should only be used when the expression ‘on the one hand’ is used above.

line 298: Delete ‘matters’.

line 523: There is no reference to Fig. 3 in the main Text of the article.

lines 537-537: Please write what ‘*’ indicates in Tab. 1.

line 550: It should be ‘RS’ instead of ‘SR’.

---

## Round 0.2 · Major Revisions

Thank you for your revisions. The reviewers indicated that there are still some issues with your manuscript that should be corrected. First, I recommend that you have a native English speaker read it before re-submission to make sure that your writing is clear and correct. Second, please address the statistical concerns of the second reviewer. Finally, please make the additional changes requested by the reviewers.

·

Basic reporting

In general, I suggest to check the correctness of the English language throughout the entire text. I have provided some corrections, however, these are incomplete.

Experimental design

No Comments

Validity of the findings

No Comments

Additional comments

Manuscript #10923

Pollinator diversity and reproductive success of Epipactis helleborine (L.) Crantz (Orchidaceae) in anthropogenic and natural habitats

Reviewer comments:

In order to improve the manuscript, suggestions are described below (line numbers are aligned with the PDF document and the Word document from the authors):

Authors have to check the correctness of the English language throughout the entire text by native linguists (speakers).

In general, I suggest that the authors use "populations in anthropogenic habitats" instead of "anthropogenic populations". In addition, use "populations in natural habitats" instead of "natural populations". Rewrite these in all occurrences in the text.

The populations of E. helleborine are not anthropogenic, they are populations that occur in human-disturbed habitats.

Abstract

PDF: Line (25), Table 1; Word document: Line (25), Table 1
I suggest to omit the word "mixed", and to write only "forest".


PDF: Line (36); Word document: Line (36)
Put the full stop (.) after the word "respectively".

Introduction

PDF: Line (54); Word document: Line (54)
Add "in western Serbia" in the sentence: In addition, Djordjević et al. (2016b) noted that Himantoglossum calcaratum, Anacamptis pyramidalis and Ophrys species often grow in habitats along the roads in western Serbia.

PDF: Line (68-69); Word document: Line (69,70)
Rewrite the sentence because there are some species that have specific deceptive mechanisms, some species have rewarding mechanisms, and there are some orchid species that are occasionally or obligatory autogamous etc.

PDF: Line (70-73); Word document: Line (72-74)
Replace and rewrite the sentence "Human disturbance with its impact on soil, moisture conditions, changes the floristic composition of plant communities and consequently directly and indirectly species structure of pollinating insects (Clemente, 2009)." after the sentence "Human disturbance, especially habitat transformation, is regarded as a principal cause of pollinator decline in global scale (Goulson, Lye & Darvill, 2008)."

PDF: Line (85); Word document: Line (86)
Write "diversity" instead of "biodiversity".

PDF: Line (85); Word document: Line (86)
Write "helleborine" instead of "helloborine".

Material and methods

Use the same subtitles "Pollinators" or "Pollinators of E. helleborine" in all sections (Materials and Methods, Results, Discussion).

PDF: Line (109-112); Word document: Line (111-113)
Rewrite the sentence as: "All shoots found in each population were measured and counted to determine population size (the total number of individuals), whereas the population area was measured using square net (Tab. 1)."

PDF: Line (113-117); Word document: Line (114-118)
Rewrite the sentence: "We made the inventory of the surrounding flora for each population of the area where it occurred, usually it was about 30 m2." I suggest to write the sentences like this in the section Study area and to omit the formula from the text and tables:

"The detailed review of the vascular flora of the studied habitats (usually the sample area was about 30 m2) is presented in Table S1. In total, 68 species of vascular plant species were recorded; 38 in natural and 38 in anthropogenic habitats. The most frequently occurring plant species in natural habitats were: Carpinus betulus, Veronica chamaedrys and Rubus sp. (registered in all study plots). The most frequently occurring plant species in anthropogenic habitats were: Achillea millefolium, Conyza canadensis, Dactylis glomerata, Galium aparine, Medicago lupulina, Poa annua, P. nemoralis and Vicia cracca. In natural habitats, 10 species of trees were recorded, whereas 5 species of trees were recorded in anthropogenic habitats (Table S1).

Information of vascular flora of the studied sites and habitats can be a part of Supplemental Information in the Table S1.

PDF: Line (129-132); Word document: Line (131-133)
Replace the sentence: "Orchids that colonize anthropogenic habitats are characterized by a specific set of features: fast growth resulting in large plant size, fast flower production and light anemochoric seeds (Forman et al., 2009)." in the Introduction section (line 61) or Discussion section.
Please, write "flower production" instead of "flowers production".

Results

In my opinion, information of the vascular flora of the studied sites and habitats can be a part of Supplemental Information (Table 2A, B). Authors can add the sentence in the section Study area that detailed review of vascular flora of studied sites is presented in Supplemental Information (for e.g., Table S1).

PDF: Line (178-187); Word document: Line (181-191)
I suggest to use these sentences in the section Study area, and to omit the text from Results because this is not one of the aims of the study, but it describes the study area.
Please, write "vascular" instead of "vascluar"; write "occurring" instead of "occuring".

Tree species are also vascular species, so the authors have to change the sentences: "In natural habitats 10 species of trees and 28 species of vascular plants were recorded. In anthropogenic habitats 5 species of trees and 33 of vascular plants were recorded. "

"The number of plant species in natural populations" can be wrongly understood and I do not recommend such a formulation.

I suggest to omit the subtitle "Flora on the populations" due to scientific incorrectness. Moreover, I suggest to write the sentences like this in the section Study area and to omit the formula from the text and tables:

"The detailed review of the vascular flora of the studied habitats (usually the sample area was about 30 m2) is presented in Table S1. In total, 68 species of vascular plant species were recorded; 38 in natural and 38 in anthropogenic habitats. The most frequently occurring plant species in natural habitats were: Carpinus betulus, Veronica chamaedrys and Rubus sp. (registered in all study plots). The most frequently occurring plant species in anthropogenic habitats were: Achillea millefolium, Conyza canadensis, Dactylis glomerata, Galium aparine, Medicago lupulina, Poa annua, P. nemoralis and Vicia cracca. In natural habitats, 10 species of trees were recorded, whereas in anthropogenic habitats 5 species of trees were recorded (Table S1).

Write "vascular flora" instead of "flora" in the title of table.

Tab. Vascular flora in natural habitats (N1-N4), B) Vascular flora in anthropogenic habitats (A1-A4)

Discussion

PDF: Line (259); Word document: Line (265)
Write "Potts et al. (2003)" instead of "Potts et. al. (2003)".

PDF: Line (259); Word document: Line (265)
Write "insect pollinators" instead of "insects pollinators".

PDF: Line (284-288); Word document: Line (290-295)
I suggest to omit or rewrite the sentence: "Our results (Fig.1, Tab.2) confirm this hypothesis as in anthropogenic habitats the most common species were small and medium-size flowering plants Achillea millefolium, Conyza canadensis, Dactylis glomerata, Galium aparine, Medicago lupulina, Poa annua, P. nemoralis, and Vicia cracca, while in natural habitats tree species dominated (mainly Carpinus betulus and Acer platanoides)."

The higher pollinator diversity in anthropogenic habitats of the studied sites may be explained by the higher insolation of anthropogenic habitats as the most common species were small and medium-size flowering plants, whereas the tree species dominated in natural habitats (Table S1). The anthropogenic habitats probably were located near the natural habitats, so the edge effects maybe also had an important role influencing the high pollinator diversity.

PDF: Line (289); Word document: Line (296)
After the first time stating the full name of the species with the authors, later in the text, use only the abbreviation: E. helleborine. The full name of the species (Epipactis helleborine) can be used also where the sentence starts with the name of the species.

PDF: Line (343); Word document: Line (352)
I suggest to rewrite the subtitle "Reproductive success and autogamy" into "Reproductive success and effect of autogamy of E. helleborine seeds".

PDF: Line (356-367); Word document: Line (365-376)

I suggest to rewrite all these sentences because they are entered completely identical as the first reviewer suggested in the first review. Since the subtitle is about reproductive success and autogamy, I suggest to replace the sentence "In addition, the edge effects may explain the higher diversity of insects in anthropogenic habitats." in the section about pollinators.

"Hágsater and Dumont (1996) noted that orchids belong to the group between ruderal and stress-tolerant plants according to Grime's (1979) theory, suggesting that they can tolerate some degree of disturbance. Since the orchid species are generally competitively weak species, a certain degree of disturbance can positively affect orchid performances, because of the reduction in the vigor of competing species by some management practices. In addition, Djordjević et al. (2016a) highlighted the ecotone characteristics of the preferred habitats of some orchid species and noted that they can tolerate a certain degree of disturbance. It is possible that E. helleborine in anthropogenic habitats has more space with a favorable light regime. Furthermore, some ecological conditions, such as soil moisture, soil pH, organic matter, could also be the reason why individuals of E. helleborine have a larger dimension in anthropogenic habitats."

PDF: Line (385-388); Word document: Line (394-397)
I suggest to connect this sentence with the conclusion of the study. The authors may write that the study confirms the general statement, or that the study is in accordance with the general statement that orchid species that are not highly specialized in relation to the type of pollinator have wider distribution and are not as rare as compared to those orchid species that have a high level of specialization (Swarts & Dixon, 2009).

In conclusion, the authors should emphasize the significance of the study, and provide the implications and suggest what would be important to investigate in the future.

Although the authors investigated anthropogenic habitats, maybe they were located near the natural habitats. Future research should explore the pollinator diversity and reproductive success of this species in the different types of anthropogenic habitats such as: the surroundings of industrial facilities, fly ash, mine tailing, highway, railway and urban parks. The research should answer the question what the level of disturbance that this species can tolerate is. Do the authors suggest the research on the potential of this species for the cultivation in horticultural purposes?

Reviewer 2 ·

Basic reporting

CLEAR, UNAMBIGUOUS, PROFESSIONAL ENGLISH LANGUAGE USED THROUGHOUT.

In general, yes. I added some suggestions for improvement under ‘General comments for the author’.


INTRO & BACKGROUND TO SHOW CONTEXT. LITERATURE WELL REFERENCED & RELEVANT.

With their revision, the authors improved this issue considerably. The literature is well references and relevant now. However, I am still missing a bit a clear focus in the introduction. I provide more detailed comments under ‘General comments for the author’.


STRUTURE CONFORMS TO PEERJ STANDARD, DISCIPLINE NORM, OR IMPROVED FOR QUALITY.

Yes.


FIGURES ARE RELEVANT, HIGH QUALITY, WELL LABELLED & DESCRIBED.

All figures are relevant and their quality is fine. In general, they are labelled and described well. The only exception is Fig. 5. In the caption of Fig. 5, it is not explained what c) and d) show. I assume a) and b) show the data for the populations in anthropogenic habitats and c) and d) the data for the populations in natural habitats. Is that correct? If so, please change the caption of Fig. 5 to something like that: ‘The dependence of reproductive success in populations in anthropogenic (a, b) and natural (c, d) habitats for the height of plants (a, c) and the number of flowers (b, d).’.


RAW DATA SUPPLIED.

No, for some data only means are shown. I still think the raw data should be supplied for a better understanding.

Experimental design

ORIGINAL PRIMARY RESEARCH WITHIN SCOPE OF THE JOURNAL.

Yes.


RESEARCH QUESTION WELL DEFINED, RELEVANT & MEANINGFUL. IT IS STATED HOW RESEARCH FILLS AN IDENTIFIED KNOWLEDGE GAP.

With their revision, the authors improved this issue.


RIGOROUS INVESTIGATION PERFORMED TO A HIGH TECHNICAL & ETHICAL STANDARD.

Yes.


METHODS DESCRIBED WITH SUFFICIENT DETAIL & INFORMATION TO REPLICATE.

The authors improved this issue considerably with their revision. However, there are still a few points that are not clear to me. In particular, I am still not sure what kind of data were exactly used for the statistical analyses and which were the exact statistical tests. For more details, please see my comments under ‘General comments for the author’.


RESEARCH CONDUCTED IN CONFORMITY WITH THE PREVAILING ETHICAL STANDARDS IN TH EFIELD

Yes.

Validity of the findings

DATA ARE ROBUST, STATISTICALLY SOUND & CONTROLLED.

No, not throughout or the authors do not explain clear enough what kind of data they analysed and what were the exact tests they use (see my comments under ‘General comments for the author’).


THE DATA ON WHICH THE CONCLUSIONS ARE BASED MUST BE PROVIDED OR MADE AVAILABLE IN AN ACCEPTABLE DISCIPLIN-SPECIFIC REPOSITORY.

Not all data are provided.


CONCLUSION WELL STATED, LINKED TO THE ORIGINAL QUESTION & LIMITED TO SUPPORTING RESULTS.

With their revision, the authors greatly solved this issue now. For a minor comment on the conclusion, please see my comment under ‘General comments for the author’.


SPECULATION IS WELCOM, BUT SHOULD BE IDENTIFIED AS SUCH.

This is fine.

Additional comments

In general, I think the authors considerably improved the manuscript – reporting on the reproductive success, pollinator community composition, and the extent of autogamy in some populations of anthropogenic and natural habitats of the orchid Epipactis helleborine in Poland – with their revision. In particular, their rearrangement of paragraphs and rewriting part of the introduction and discussion made the manuscript much easier to read and understand. However, I still have some concerns regarding the statistical analyses (maybe that is just related to an unclear description of the statistical analyses), and, in the introduction, I still miss a clear development of the research questions. Below, I list my concerns more detailed. Whenever I mention line numbers, I refer to the line numbers in the PDF file.


ABSTRACT

lines 19+20: Do not use a line break.

lines 20+21: This sentence does not seem to be grammatically sound. Maybe replace ‘As it is suggested orchids colonizing…’ by ‘It is suggested that orchids colonizing…’.

line 21: It should be ‘e.g.’ instead of ‘eg.’ and ‘flower production’ instead of ‘flowers production’.

lines 21-23: Here, I miss WHY you want to compare pollinator diversity and reproductive success of E. helleborine between natural and anthropogenic habitat types. Maybe write something like ‘However, it is not well known how pollinator diversity and reproductive success of E. helleborine differs in populations in anthropogenic habitats compared to populations in natural habitats.’. (The other reviewer suggested using the term ‘populations [of E. helleborine] in anthropogenic/natural habitats’, which, I think, is appropriate, and the authors should use it consistently throughout their manuscript (is not yet the case).

lines 26-29: Write all numbers as words.

lines 27+28: The traits measured are grammatically not consistently used (mixture of singular and plural as well as definite and indefinite). My suggestion is: ‘…height of plants, length of inflorescences, as well as numbers of …’.

lines 31+32: Add a comma as follows: ‘In both types of populations, the main …’.

line 33: ‘With respect to’ is more appropriate than ‘According to’.

lines 33+34: This sentence is grammatically not sound. My suggestion is: ‘According to the type of the pollinators’ mouthpart, chewing (39%), sponging (34%) and chewing-sucking (20%) pollinators prevailed in anthropogenic habitats.’.

lines 34+35: Add a comma as follows: ‘In natural habitats, pollinators …’.

lines 35+36: There are three types of insects (1) chewing-sucking, (2) piercing, and (3) sucking but only two values of percentages. Please specify.

lines 38+39: The authors argue here that a ‘higher number of visits by pollinators and their greater species diversity but also … the bigger size of plants’ might play an important role. However, these differences are not mentioned in the paragraph ‘Results’ in the abstract. They should be added, if used as argument.

line 41: Add a sentence what these results imply on the ability of E. helleborine in using anthropogenic altered habitats and in colonising America. Because I think that is a key point of the manuscript.


INTRODUCTION

With their revision, the authors improved the readability and the focus towards their research questions. However, I am still concerned about two major points.

First, I still think it is not clear, how the paragraph on orchid population biology (starting at line 60), relates to the question of difference between populations in anthropogenic and in natural habitats WITHIN one orchid species. To my knowledge, all listed aspects apply to orchids in general, with all aspects probably being disadvantageous for a plants ability to colonise anthropogenic habitats. Thus, the question would rather be: why and how do some orchid species nevertheless manage to colonise anthropogenic habitats. Please clarify that.

Second, the authors explained now the relevance of pollinators with different mouth parts. However, the arguments are made at the among-species level and not at the within-species level at which this study was conducted. Thus, I am still unsure what the authors’ purpose of looking at pollinator mouth parts is. Do they predict/know that flower morphology in E. helleborine differs between populations in anthropogenic and natural populations and thus want to test whether this is correlated with the type of the pollinators’ mouthpart in these two habitat types? Or is nothing/little known about the diversity of mouth part types of the pollinators of E. helleborine in general? Please specify. Mentioning this point in a sentence in the introduction would be helpful for a better understanding.

In addition, I have some additional, more minor comments on the introduction. They are detailed here.

lines 47+48: Do not use a line break because a single sentence (lines 45-47) cannot be a paragraph.

lines 48: What is the meaning of the sentence ‘Shrinking natural habitats, transformation is causing …’? Do you mean ‘Shrinking natural habitats and anthropogenically induced habitat transformation is causing …’?

lines 48-50: I think it is contradictory to say in the first sentence that ‘many’ orchid species extinct and in the following sentence that ‘many’ species can use anthropogenic habitats as analogues of natural habitats. Would it not be more appropriate to rewrite the second sentence as follows: ‘However, some orchid species, especially in …’?

line 51: It should be ‘A recent study’ instead of ‘The recent study’.

line 57: Add a comma after ‘Moreover’.

line 60: This sentence is grammatically not sound. Rewrite it e.g. as follows: ‘An important aspect of population biology in orchids is their unique reproductive system…’.

lines 61-63: Also this sentence somehow does not work out grammatically. Rewrite it e.g. as follows: ‘They are characterized by a long life span (up to 15 years), they rely on the presence of mycorrhizal symbionts, and they have a highly specialized mechanism of insemination…’.

line 63: It should be ‘results from’ instead of ‘results of’.

lines 68-70: As the present study is conducted in rewarding orchid and the authors do not refer on deceptive orchids later in their manuscript, I think the two sentences ‘Some orchid species developed specific deceptive and rewarding mechanisms. Moreover the studies showed that reproductive success in deceptive orchids is lower than that in rewarding ones (Kindlmann & Jersakova, 2006).’ distract from the main focus of the manuscript and should be omitted. Maybe you could replace them with a sentence like: ‘More than 60% of all orchid species are pollinated by a single pollinator (Tremblay, 1992; Tremblay et al., 2005).’
[Tremblay, R.L. 1992. Trends in the pollination ecology of the Orchidaceae: evolution and systematics. Canadian Journal of Botany 70: 642-650.
Tremblay, R.L., Ackerman, J.D., Zimmermann, J.K. and Calvo, R.N. 2005. Variation in sexual reproduction in orchids and its evolutionary consequences: a spasmodic journey to diversification. Biol. J. Linn. Soc. 84: 1-54.]

lines 70-72: What is the meaning of the sentence ‘Human disturbance, on its impact on soil, moisture conditions, changes the floristic composition of plant communities …’? Do you mean ‘Human disturbance, in particular its impact on soil moisture conditions, changes the floristic composition of plant communities …’?

line 76: Autogamy might also require visitations by pollinators. Do you mean autonomous selfing? Please specify.

line 80: Add a hyphen as follows: ‘auto- and allogamy’.

line 85: It should be ‘in relation to’ instead of ‘according to’.

line 86: Replace ‘aspect’ by ‘information’.

line 88: The expression ‘On the other hand’ is only suitable when earlier in the paragraph the expression ‘on the one hand’ is used. Replace the expression by e.g. ‘in addition’.

line 89: Write the numbers in words, i.e. ‘…from one or two morphologically…’. It should be ‘…flowers, which means…’ instead of ‘…flower what means…’


MATERIALS AND METHODS

With their revision, the author considerably improved the understandability of the section ‘Materials and Methods’. However, a few points are still unclear, in particular concerning the statistical analyses, and are detailed here.

lines 99-132: The understandability of the how the study was conducted and why it was conducted the way it was would be improved, if the ‘Materials and Methods’ section was started with the information on the study species (i.e. line 118-132). The title (line 99) should then be changed to ‘Study species and study are’. Alternatively, if the authors have a specific reason why they first describe the study area, please elaborate on this.

lines 102-109: In their reply to the reviewers comments, the authors nicely describe more detailed differences between populations in anthropogenic habitats and natural habitats. This information would be helpful in the text. Particularly informative is the fact that the populations in anthropogenic habitats were located in/near cities/villages. This could nicely been shown on a map and combined with the photos of Fig. 1.

line 109: Do not only refer to Fig. 1 but also to Tab. 1. Does ‘Faliński, 2001’ refer to the whole sentence (then it is fine) or only to ‘Galio sylvatici-Carpinetum betuli Oberd. 1957 in the Strict Reserve Białowieża Primeval Forest’? In the latter case, it should be cited directly after ‘…Primeval Forest’ (i.e. before the reference to Fig. 1 and Tab. 1).

line 110: What was measured? Please specify.

line 111: Is population size the number of individuals or the number of shoots? Please be precise/consistent in the text.

lines 111+112: How large was this square net and how was it laid out? Please specify.

line 113: It should be ‘We made an inventory of the surrounding flora…’. What is meant by ‘for each populations of area it occurred’? Do you mean ‘for each population in the area E. helleborine occurred’? Please specify.

line 114: Usually it was 30 m2. Please indicate how many m2 it was in the other cases.

line 115: A verb is missing after ‘occurrence’ (e.g. was calculated).

line 120: It should be ‘It usually grows …’.

line 131: It should be ‘flower production’. To my knowledge, all orchids have light, anemochoric seeds. Thus, it is not a specific feature and should be omitted.

line 137: Refer to Fig. 1 and Tab. 1.

lines 137+138: Do not use a line break because a single sentence (lines 134-137) cannot be a paragraph.

line 144: What is the exact meaning of the sentence ‘Invertebrate specimens were collected until the transfer of pollinia by pollinators were [should be ‘was’!] observed.’? Does this mean that between 9am and 7pm all invertebrates found on E. helleborine flowers/inflorescences were collected and in the end a great many of them were discarded when no pollinia transfer was observed? Were really all kind of invertebrates collected (i.e. also snails…) or only insects/arthropods. Please describe these issues more precisely in the text.

line 148: Replace ‘talked about’ by ‘corresponded to’.

line 149: Change as follows: ‘…Bombus species, for which only the number of visits was counted’.

lines 153-159: This information has not related to how reproductive success was measured. Thus, move it to and include it in the sentence in lines 109-112. Add instead a sentence with information on the date when reproductive success was counted and how in the paragraph ‘reproductive success’.

line 155: Why was the number of juvenile shoots measured? It is not mentioned anymore later in the manuscript. Or was population size (total number of shoots in a population) measured as the sum of juvenile shoots and the number of flowering shoots? Please specify that in the text.

line 162: It should be ‘the autogamy experiment’.

lines 162-164: For clarity, change the order of the sentences as follows (otherwise it is confusing why there were no bags in the control plants): ‘… the autogamy experiment. Flowers on each shoot were counted and inflorescences were covered by bags made from mosquito net. We used also ten control plants (not covered by mosquito-net bags). After three months …’.

line 165: It should be ‘the number of fruits set’. Omit the reference to Tab. 1 as the information is not shown there.

lines 171-176: In their response to the reviewers comments, the authors write that they now used the data for all individuals instead of means. However, they do not describe, whether they then controlled for population identity, i.e. whether they included population as random factor in their models, because individuals of a population are not independent data points. Please describe in the manuscript exactly for which analyses you used means and for which analyses you used values of individuals. In the latter case also describe what kind of model you used.

line 172: Add a comma as follows: ‘between habitats, we used’.


RESULTS

The rearrangement of paragraphs the authors conducted in their revision considerably facilitated the understandability of their findings. I have a few minor comments as detailed here.

line 179: Omit ‘on analyses habitats,’ or better explain what it means.

lines 180+181: In general, the authors first refer to the populations of anthropogenic habitats and then to populations of natural habitats. To be consistent and thus make it easier to read your text, change the sentence as follows: ‘The number of plant species in populations of anthropogenic habitats ranged from 16 to 27 species, whereas it ranged from 16 to 25 in populations of natural habitats (Tab. 2A,B).’.

line 189: ‘Epipactis helleborine’ should be abbreviated as ‘E. helleborins’.

line 193: Change as follows: ‘In populations of natural habitats, the most frequent …’.

lines 195-198: As I understand ‘the Białowieża Primeval Forest’ is the population in the natural habitat and it would be much easier if it would be just referred to as such. Thus, the sentence should be reformulated e.g. as follows: ‘In both types of habitats, Diptera and Hymenoptera clearly predominated, with 41% and 52% of all the pollinators observed in populations of anthropogenic habitats, and with 59% and 37% observed in the population of the natural habitat…’.

line 198: It should be Fig. 3 instead of Fig. 4.

line 199: It should be ‘in the population of the natural habitat’.

lines 200+201: According to Fig. 3, these occasional pollinators also occur in only one of the two habitat types. It would be interesting to include this information in the text, e.g. as follows: ‘Occasionally, single individuals of grasshoppers (Orthoptera) and earwigs (Dermaptera) were also noted as pollinators of E. helleborine in populations of anthropogenic habitats and scorpion flies (Mecoptera) in the population of the natural habitat.’

line 208: To be consistent change as follows: ‘(Formicidae – 16% and …’. Add a reference to Fig. 3 at the end of the sentence.

lines 208+209: Replace ‘the Białowieża Primeval Forest’ by ‘the population of the natural habitat’. Do that throughout the manuscript.

line 211: To be consistent change as follows: ‘(Vespidae – 58% and …’.

line 212: To be consistent change as follows: ‘(Apidae – 24% and …’.

line 213: To be consistent change as follows: ‘(Formicidae – 17% and …’. Add a reference to Fig. 3 at the end of the sentence.

line 215: To be consistent change as follows: ‘…Culicidae – 44% …’.

line 217: To be consistent change as follows: ‘…Mecoptera – 36%)’ and ‘(Apidae – 14%)’.

line 218: To be consistent change as follows: ‘(Cuculidae – 6%)’.

line 219: It should be ‘In the population of the natural habitat, the main…’.

line 228: Why are there two p values? Is one (p = 0.02) the test statistics t?

lines 228-230: To be consistent change as follows: ‘In populations of anthropogenic habitats, reproductive success ranged from 77.8% (A4 - in 2011) to 100% (A2 - 2012), while in populations of natural populations it ranged from 44.4% (N1 - 2011) to 85.0.3% (N1 -2012) (Tab. 4)’.

lines 230-232: What is the statistics of the difference in the number of flowers? Please add this information. In their response to the reviewers comment, the authors that they measured flowers and shoots but want to publish it somewhere else. However, some of these data are still mentioned in the sections ‘Materials and Methods’ and ‘Results’ and shown in Tables. Thus if you want to publish it somewhere else, omit all these information and data completely from the manuscript.

line 232: It should be ‘ranged from’.

line 235: Add a comma after ‘populations’.

line 241: Replace ‘autogamy’ by ‘the autogamy treatment’.

line 243: According to the information in ‘Materials and Methods’ it should be 30 individuals. What happened to the rest and in how many individuals are there in which habitat type? Please specify. Why are there two p-values. Please double-check and only show the correct one.

lines 245+246: The statistics should be included at the end of the sentence before the reference to Tab. 5.

line 246: It should be ‘flowers per inflorescence’.

line 252: ‘proportion’ is more appropriate then ‘amount’.

line 253: It should be: ‘and was higher compared to…’.


DISCUSSION

Again, I think the authors’ revision considerably improved the discussion. Here are some minor comments.

lines 258-261: Change as follows: ‘According to Kearns, Inouye & Waser (1998) and Potts et. al. (2003), in approximately 90% of all angiosperm species, insect pollinators are involved in sexual reproduction of these plants. In orchids, approximately 70% of the species are closely related to specific insect pollinators (Neiland & Wilcock, 1998).

line 268: It should be ‘Regarding’ instead of ‘According to’. Add a comma after ‘insect groups’.

line 274: Write ‘those latter observation’ to make it more clear.

line 276: It should be ‘can be an explanation of such a phenomenon’.

line 291: Add a comma after ‘literature’.

line 310: Add a comma after ‘However’.

line 319+320: As more than one Epipactis species has been investigated, change e.g. as follows: ‘Similar patterns were also observed in other species of Epipactis such as E. palustris…’.

line 321: Add a comma before ‘and’.

line 332: Add a comma before ‘who’.

line 333+334: Use the Greek letters alpha and beta instead of a and b, respectively.

line 339: More common than what? Maybe add e.g. ‘than known up to now’.

line 339+340: Do not use a line break.

line 340-342: These sentence is grammatically not sound. Change it e.g. as follows: ‘Overall, these results indicate that E. helleborine has a diverse group of pollinators, which may promote the genre in very rapidly changing areas transformed by man and which is one of the key features of apophytes.’.

line 348: I think ‘proved’ is a bit too strong. Maybe change it to ‘provide evidence’.

lines 350-352: This sentence is grammatically not sound. Change it e.g. as follows: ‘Despite some differences in the number of fruits between populations of the two habitats, we found no significant differences in the number of fruits formed between autogamy and natural pollination, which is congruent with work of Weijer (1952).’.

line 353: It should be ‘in the life cycle’.

lines 353+354: What is the meaning of the sentence ‘Fruits were produced in both types of habitats.’? Omit it or explain more precisely.

line 355: Use ‘On the one hand’ may only be used when ‘on the other hand’ is used later on.

lines 377+378: How does this simultaneous opening of flowers relate to the present study? Please elaborate or omit this sentence.

line 381: It should be ‘Ågren’ instead of ‘Agren’. Change that also in the ‘References’ section.

line 386+387: Specify and change as follows: ‘…have wider distribution ranges and are less rare than orchid species that have a high level of pollinator specialization…’.

line 389: Change as follows: ‘…habitats, which might be caused…’.

line 390: Add a comma after ‘Moreover’.

line 392-394: To close the circle of the story, and reinforce a key point of your study (at least when reading your manuscript I see it as a key point) include a sentence/phrase that your study contributes to a better understanding why E. helleborine is one of the (few) orchid species that manages to use anthropogenic habitats.


FIGURES

line 607: It should be ‘…bordering in a (or the?) Pinus…’.

Fig. 5: Use the same scale for all graphs. Indicate in the figure which graphs are from the populations of anthropogenic habitats and which from populations of natural habitats. Indicate in the graphs which regressions were significant. Only draw a regression line when it is significant.

line 627: What is c) and d)? Please specify.


TABLES

Tab. 1: Move the columns ‘Coordinates’ and ‘Altitude’ after the column ‘Locality’.

Tab. 2: To be consistent with the order in Fig. 1 and Tab. 1, first show the table of the anthropogenic habitats and then the table of the natural habitats. To be more precise, write ‘Total number of species’ instead of ‘Total’.

Tab. 4: Also include the values of the standard deviation in the columns ‘HP’ and ‘RS’.

Tab. 5: Include the values of the standard deviation. Moreover, I think it would be more informative to show these results as correlation plots instead of in a table.

Tab. 6: Include the values of the standard deviation.

---

## Round 0.3 · Minor Revisions

Thank you very much for your careful and labor-intensive revision.

There are some additional changes I would like you to make so that the language will flow better. These are listed below:

1) line 34, "were larger" instead of "are bigger"
2) line 43, "the larger" instead of "a bigger"
3) line 47, "has allowed" instead of "has caused"
4) line 57, "found anthropogenic habitats analogue to natural ones" does not make sense. Do you mean they respond to these habitats similarly to natural ones?
5) line 62, "can be" instead of "was determined as"
6) line 65, remove "have" in front of "noted"
7) line 68, surface soil layers
8) lines 69-70, renovation or construction of roads; the rest of the sentence does not make sense
9) line 77, depending on the species
10) line 78, reproductive success
11) line 79, question arises as to which
12) line 92, furthermore, has been
13) line 95, relation to certain types
14) line 96, there are no detailed data
15) line 97, about the diversity
16) line 99, flowers;
17) line 102, they can only be
18) line 118, usually grows
19) line 121 and mine tailings
20) line 124, species with a broad
21) line 125, what does restated mean?
22) line 135, number of fruits
23), line 140, net was approximately
24) line 149, refer to map after Sulejow
25) line187, provided according to Gilliot (2005)
26) line 189, shoots in the
27) line 203-204, "determined the function of linear regression model" does not make sense, Please rephrase.
28) line 214, populations from the natural habitat (NOTE: PLEASE USE THIS THROUGHOUT)
29) line 218, populations from the anthropogenic habitats (PLEASE USE THIS THROUGHOUT)
30) line 277, in the inflorescence
31) line 282, (which varied from...)
32) line 291, is still fragmentary (Aizen & Vazquez, 2006);it is suggested
33) line 357, as was shown
34) line 364, who noted
35) line 371, than was shown
36) line 372, it is not clear what you mean by genre here. Do you mean the genus? If so, include the genus name.
37) line 387, According to Grime's
38) line 388, disturbances. Hagsater and Dumont
39) line 391, against the general
40) line 394, This does not make sense. Ecotone characteristics cannot point out something. Please rephrase.
41) line 406, habitats we studied
42) line 409, correlation between density of plants and what? This does not make sense; you need to say what the correlation is between.
43) line 414, wider than has been suggested
44) line 416, which might have been
45) line 418, autogamy was not uncommon as a reproductive strategy
46) line 419, we found no
47) line 421, species that has been
48) line 426, "contributes to a better understanding of" is repetitious. Use "helps to explain" instead.
49) line 428, manner comparable to that of natural ones...The question of how...
50) line 429-30, To answer the question, we plan further research on....

Thank you, and I look forward to seeing your new revision.

---

## Round 0.4 · Minor Revisions

Thank you so much for making the large number of changes I requested. The paper is almost ready, but I need you to do a bit more work on it. On line 206, you say that you used a "function of linear regression model" to estimate correlations. It is not clear what you mean by this. Please give more details so the reader can understand what you did. Another minor change is on line 143. The phrase should be "approximately 1 m2" (remove extra words between "approximately" and "1."

---

## Round 0.5 · accepted · Accept

Thank you so much for clarifying your regression model, and for your patience in making all of the changes I have requested. Again, congratulations!